# Reward signaling in a recurrent circuit of dopaminergic neurons and peptidergic Kenyon cells

Radostina Lyutova [1], Mareike Selcho [1], Maximilian Pfeuffer[1], Dennis Segebarth [1,2], Jens Habenstein [1,3], Astrid Rohwedder[4], Felix Frantzmann [1], Christian Wegener [1], Andreas S. Thum [4] & Dennis Pauls [1]

Dopaminergic neurons in the brain of the *Drosophila* larva play a key role in mediating reward information to the mushroom bodies during appetitive olfactory learning and memory. Using optogenetic activation of Kenyon cells we provide evidence that recurrent signaling exists between Kenyon cells and dopaminergic neurons of the primary protocerebral anterior (pPAM) cluster. Optogenetic activation of Kenyon cells paired with odor stimulation is sufficient to induce appetitive memory. Simultaneous impairment of the dopaminergic pPAM neurons abolishes appetitive memory expression. Thus, we argue that dopaminergic pPAM neurons mediate reward information to the Kenyon cells, and in turn receive feedback from Kenyon cells. We further show that this feedback signaling is dependent on short neuro-peptide F, but not on acetylcholine known to be important for odor-shock memories in adult flies. Our data suggest that recurrent signaling routes within the larval mushroom body circuitry may represent a mechanism subserving memory stabilization.

[1] Neurobiology and Genetics, Theodor-Boveri Institute, Biocenter, University of Würzburg, D-97074 Würzburg, Germany. [2] Institute of Clinical Neurobiology, University Hospital of Würzburg, D-97078 Würzburg, Germany. [3] Department of Behavioral Physiology and Sociobiology, Theodor-Boveri Institute, Biocenter, University of Würzburg, D-97074 Würzburg, Germany. [4] Department of Genetics, University of Leipzig, D-04103 Leipzig, Germany. Correspondence and requests for materials should be addressed to D.P. (email: dennis.pauls@uni-wuerzburg.de)

Memory can be defined as a change in behavior due to experience. It enables animals and humans to adapt to a variable environment. To do so, the current situation must be continuously re-evaluated and recorded in the brain. On the neuronal level, memories are encoded as changes in activity or connectivity which outlast the triggering environmental stimulus. Thus, the storage of relevant information in a memory trace is a dynamic, multi-dimensional process. Such complex calculations require a neuronal network with feed-forward and feedback motifs enabling the system to integrate current information, to provide learned information, and to organize decision making for the behavioral outcome based on an integration of all information.

In *Drosophila*, the mushroom bodies (MBs) represent a multimodal integration center incorporating a variety of different sensory stimuli. A major function of the MBs is to establish, consolidate and recall associative odor memories, in both larval and adult *Drosophila*[1–7]. During classical olfactory conditioning, odor information (CS, conditioned stimulus) is represented by an odor-specific subset of third-order olfactory neurons, the MB intrinsic Kenyon cells (KCs). KCs also receive information about reward or punishment (US, unconditioned stimulus) mediated by octopaminergic neurons (OANs) and/or dopaminergic neurons (DANs). The coincidental odor stimulation of KCs and MB input neuron (MBIN; OANs or DANs) stimulation via rewarding or punishing stimuli leads to memory formation[2,4,5,8–10].

The larval MB consists of eleven distinct compartments which are defined by the innervation pattern of MBINs and MB output neurons (MBONs)[11,12]. The four compartments of the MB medial lobe are each innervated by a single DAN of the pPAM (primary protocerebral anterior medial) cluster. These dopaminergic MBINs are essential to mediate the internal reward signal to the medial lobes of the MBs during odor-sugar learning[12–14].

The current assumption is that within the MBs synaptic plasticity occurs and is transferred into a conditional behavioral output driven by MBONs and downstream premotor circuits[2,4,5]. Recently, Eichler and coworkers reconstructed the connectome of the larval MBs providing a complete neuronal and synaptic circuit map of KCs, MBINs, and MBONs on the electron-microscopic level[11]. This study identified a canonical circuit motif in each compartment relying on already well-described synaptic connections—MBINs to KCs and KCs to MBONs. Interestingly, the study also revealed hitherto undescribed synaptic connections. So far, the function of these new connections is largely unexplored.

It is, however, tempting to speculate that feedback signaling within the MB circuitry allows modulation of neuronal activity on different levels. This is supported by recent findings in adult *Drosophila*, where feedback signaling to MBINs is important for appetitive olfactory long-term memory formation[15,16].

Here, we experimentally address larval KC-to-dopaminergic MBINs signaling involved in appetitive olfactory learning. We find that optogenetic activation of larval MB KCs paired with odor stimulation induces an appetitive memory, which is dependent on KC-to-DAN signaling. Our data further suggest that feedback signaling during reward learning depends on short neuropeptide F, but not acetylcholine, both signals produced and released by KCs[17,18]. Moreover, we show via calcium imaging that dopaminergic pPAM neurons specifically respond with an increase in Ca$^{2+}$ levels to MB KC activation. Our results indicate a novel peptidergic feedback mechanism between MB KCs and dopaminergic MBINs that is potentially involved in stabilizing odor-reward memories in the *Drosophila* larva.

## Results

**Optogenetic activation of KCs induces memory formation.** We used a well-established 1-odor reciprocal training regime to analyze classical olfactory learning and memory in *Drosophila* larvae[19,20]. To address whether optogenetic activation of MB KCs affects learning and memory, we used the Gal4/UAS system[21] and the *H24-Gal4* driver line[22] to express Channelrhodopsin (UAS-ChR2-XXL; ref. [23]). *H24-Gal4* specifically expresses Gal4 in almost all KCs and only a few neurons located in the ventral nervous system (VNS) (Fig. 1b).

In contrast to the standard procedure, olfactory stimuli were paired with a blue light-dependent activation of KCs (KC-substitution learning) (Fig. 1c) as an artificial substitute for a conventional US like sugar or salt[5,19]. Strikingly, experimental larvae showed a significant appetitive memory in contrast to genetic controls that did not show any learning performance (Fig. 1a, Supplementary Fig. 1). This result suggests that the artificial activation of KCs is sufficient to induce an appetitive memory. To validate our findings and to exclude that the effect is based on the activation of cells outside the MBs in *H24 > ChR2-XXL* larvae, we additionally tested different MB driver lines: *OK107-Gal4* (expression in all KCs; ref. [24]) (Fig. 1e), *201y-Gal4* (315 KCs; ref. [3]) (Supplementary Fig. 1), and *MB247-Gal4* (341 KCs; ref. [3]) (Supplementary Fig. 1). In all three cases artificial activation of KCs led to appetitive memory formation in experimental larvae (Fig. 1d, Supplementary Fig. 1). Thus, we conclude that optogenetic activation of KCs is sufficient to induce an appetitive memory, as the different driver lines only overlap in the MBs. We suggest that the lack of aversive olfactory memory expression upon KC activation is based on the test situation in our KC-substitution learning experiment (Fig. 1c, Supplementary Fig. 2). *Drosophila* larvae are known to express aversive memories only in the presence of the aversive US in the test situation[5,19,25,26]. In contrast, to recall appetitive memories the appetitive US must be absent during the test (Supplementary Fig. 2). Since in our experiments larvae are tested on pure agarose, they will expect a significant gain through the expression of appetitive memory (Supplementary Fig. 2). To test whether the artificially induced memory is specific, we challenged *H24 > ChR2-XXL* larvae on a fructose and amino acid test plate[26]. While the presence of fructose during the test abolished memory retrieval, appetitive memory expression was unaffected in experimental larvae tested on amino acid (Fig. 1g, i), suggesting that ChR2-dependent broad activation of KCs induces the expression of a specific sugar memory, rather than an unspecific reward memory.

**KC activation does not affect locomotion and odor responses.** The MBs integrate a variety of different sensory modalities. To exclude that optogenetic activation of KCs alters general behaviors like locomotion or processing of sensory stimuli necessary for classical olfactory learning (i.e., odors), larvae were assayed for (i) locomotion and (ii) innate odor preference. We analyzed locomotion using the FIM tracking system[27]. For this, larvae were monitored under red light for 1 min. Subsequently, we exposed larvae to blue light and recorded again for 1 min (Fig. 2a–c). Optogenetic activation of KCs did not affect the velocity and crawled distance over time, suggesting that artificial activation of KCs does not affect general locomotion parameters (Fig. 2a–c, Supplementary Fig. 3). However, experimental larvae showed a significant decrease in the number of turns and stops under blue light exposure, suggesting a change in orientation behavior (Supplementary Fig. 3). Next, we tested whether optogenetic activation of KCs alters innate odor preference, which would perturb the conditioning experiments of our study. For this,

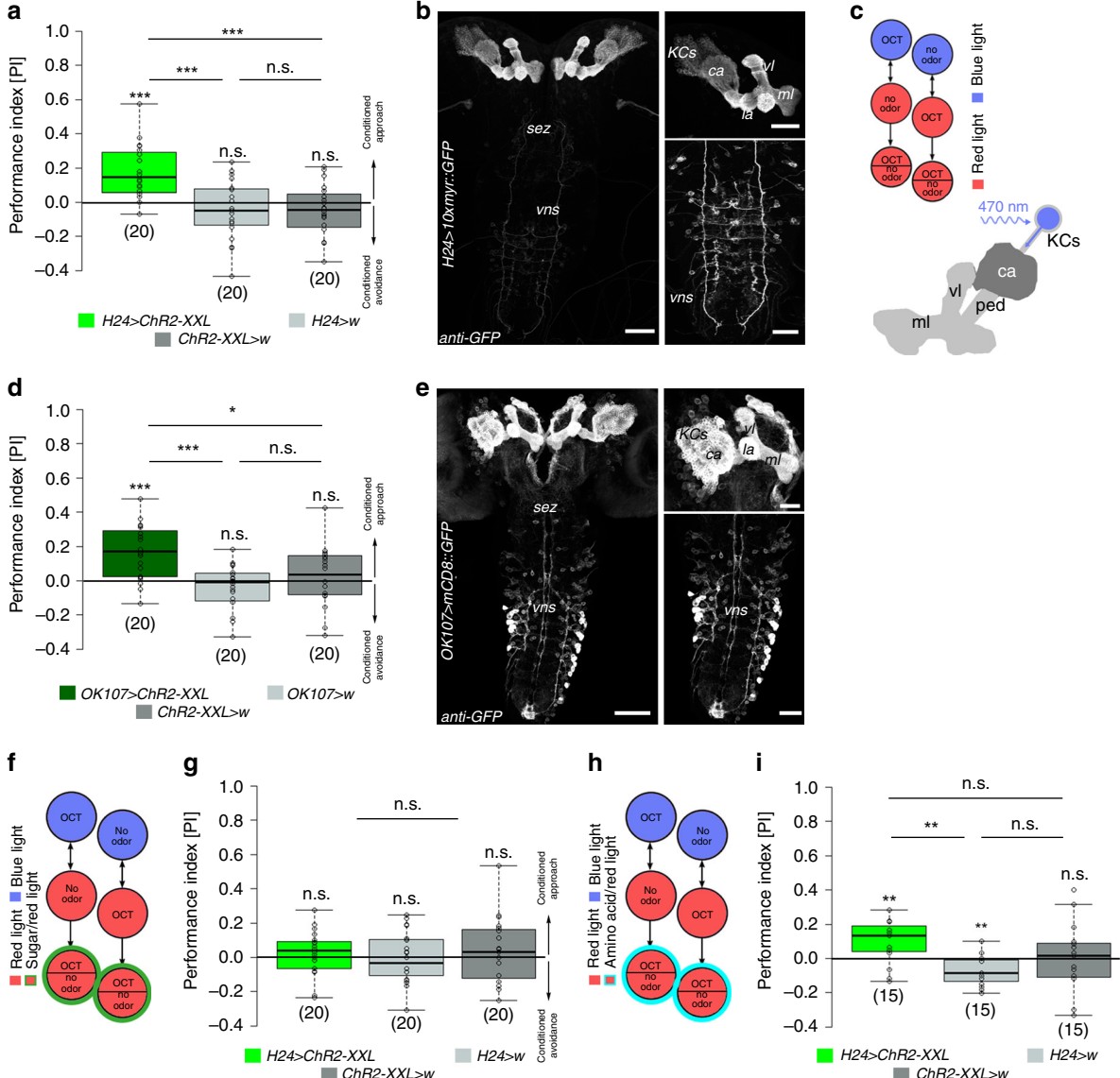

**Fig. 1** Optogenetic activation of KCs induces reward memory formation. **a** Conditional optogenetic activation of KCs coupled to odor information during training is sufficient to induce an appetitive memory using *H24-Gal4*. **b** Expression pattern of *H24-Gal4* crossed with 10xUAS-*myr::GFP* stained with anti-GFP (white). *H24-Gal4* shows expression in almost the complete set of MB KCs and a few additional cells of the vns. **c** Illustration of the 1-odor reciprocal training regime used in this study (substitution learning) and a schematic drawing of the larval MB expressing ChR2-XXL in KCs. **d** Conditional optogenetic activation of KCs coupled to odor stimulation during training is sufficient to induce appetitive memory expression (*OK107 > ChR2-XXL*). **e** Expression pattern of *OK107 > mCD8::GFP* stained with anti-GFP (white). **f** Schematic drawing of KC-substitution learning using a sugar test plate (green circle). **g** Appetitive memory expression in *H24 > ChR2-XXL* larvae is abolished during KC-substitution learning when larvae are tested on a sugar plate. **h** Schematic drawing of KC-substitution learning using an amino acid test plate (cyan circle). **i** Appetitive memory expression in *H24 > ChR2-XXL* larvae is unaffected during KC-substitution learning when larvae are tested on an amino acid test plate. ca: calyx; ChR2-XXL: channelrhodopsin2-XXL; KCs: Kenyon cells; la: lateral appendix; ml: medial lobe; 10xmyr::GFP: 10xUAS-myristoylated green-fluorescent protein; OCT: octanol; ped: peduncle; sez: subesophageal zone; vl: vertical lobe; vns: ventral nervous system; w: w[1118]. The number below the box plots refers to the *N* number. A pairwise Student's *t* test or pairwise Wilcoxon test (both including Bonferroni-Holm correction) was used. Significance levels: n.s.: *p* > 0.05, *p* < 0.05, **p* < 0.01, ***p* < 0.001. Scale bars: 50 μm and 25 μm for higher magnifications. Data are mainly presented as box plots, with 50% of the values of a given genotype being located within the box, and whiskers represent the entire set of data. No data were excluded. Outliers are indicated as open circles. The median performance index is indicated as a thick line within the box plot

larvae were assayed in a simple choice test using pure octanol (OCT; the odor used in our KC-substitution learning) and diluted amylacetate (AM; commonly used for the two-odor reciprocal experimental design; refs. [5,19]), respectively. Optogenetic activation of KCs in experimental larvae did not change innate preference to OCT and AM (Fig. 2d, e). To verify that the activation of all KCs does not change odor quality to an unspecific attractive odor, larvae were tested in a choice test with exposure to OCT

and AM opposing each other. Previous studies showed that pure OCT and 1:40 diluted AM are balanced in their attractiveness, while larvae prefer pure AM over pure OCT. As expected, experimental larvae showed a significant approach toward AM, which was indistinguishable from controls (Fig. 2f); coherently larvae were randomly distributed in the choice test between OCT and 1:40 diluted AM (Fig. 2g). These results suggest that an optogenetic activation of KCs does not change innate odor

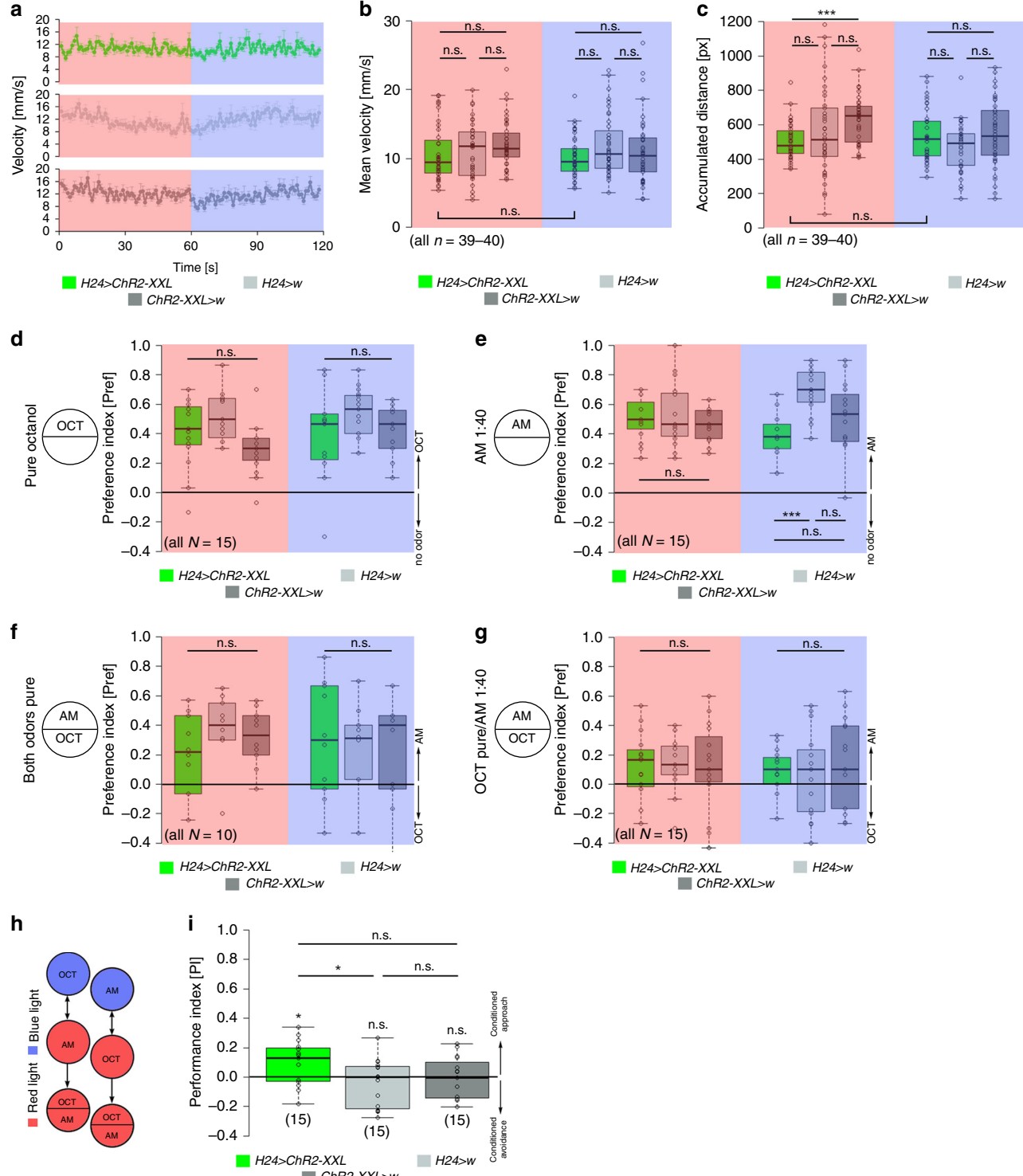

**Fig. 2** Optogenetic activation of KCs does not alter locomotion or odor responses. **a–c** Conditional optogenetic activation of KCs does not change larval velocity (**a**, **b**) and accumulated distance (**c**). The light regime is indicated by the red and blue rectangles: larvae were monitored under red light for 1 min, subsequently under blue light for another minute. **d–g** Similarly, innate odor preference to OCT (**d**) and diluted AM (**e**) is not altered due to conditional optogenetic activation of KCs. In line, experimental larvae showed normal performance in an odor discrimination task using OCT and either pure AM (**f**) or 1:40 diluted AM (**g**). **h**, **i** Conditional optogenetic activation of KCs is sufficient to induce an appetitive memory using a 2-odor reciprocal training regime indicating that experimental larvae can distinguish different odors despite artificially activated KCs. AM: amylacetate; ChR2-XXL: channelrhodopsin2-XXL; KCs: Kenyon cells; OCT: octanol; w: w[1118]. The number below the box plots refers to the N number. A pairwise Student's t test or pairwise Wilcoxon test (both including Bonferroni-Holm correction) was used. Significance levels: [n.s.]$p > 0.05$, *$p < 0.05$, **$p < 0.01$, ***$p < 0.001$. Error bars (**a**) represent the standard error of the mean. Data are mainly presented as box plots, with 50% of the values of a given genotype being located within the box, and whiskers represent the entire set of data. No data were excluded. Outliers are indicated as open circles. The median performance index is indicated as a thick line within the box plot

preferences. We also repeated the KC-substitution learning experiment using *H24-Gal4* and *OK107-Gal4* in the two-odor reciprocal design (Fig. 2h; refs. [5,19]). Similar to our initial results, optogenetic activation of KCs was sufficient to induce appetitive memory, verifying that odor discrimination (at least between AM and OCT) is not disturbed (Fig. 2i, Supplementary Fig. 4). Taken together, these results suggest that optogenetic activation of KCs does not perturb locomotion and innate preference to odors essential for KC-substitution learning in our study.

**KC activation specifically drives appetitive memory expression.**
Our results illustrate that artificial activation of KCs paired with odor exposure is sufficient to specifically induce appetitive memory. Therefore, we tested whether the optogenetic activation of KCs mimics an internal reward signal that is sufficient to induce an associative reward memory in the KC-substitution experiment. For this, we tested larvae in a simple light preference test (Fig. 3a). Feeding larvae strongly avoid light to ensure staying within the food[28–30]. Thus, we challenged *H24 > ChR2-XXL* larvae to choose between blue light and darkness. Given the idea that the activation of KCs induces internal reward signaling, we assumed that control larvae would prefer darkness over blue light, whereas *H24 > ChR2-XXL* larvae would show a reduced dark preference as they would experience blue light as reward. As expected, control larvae significantly avoided illuminated rectangles after 1 and 3 min. In contrast, experimental larvae showed light avoidance only after 3 min, but were randomly distributed after 1 min suggesting a delayed light avoidance (Fig. 3b). To verify that larvae are challenged to decide between reward and innate light avoidance in our assay, we repeated the experiment, but this time used red light (Fig. 3c), which does not elicit innate light avoidance in the larva[31,32], but activates KCs in experimental larvae as we expressed UAS-Chrimson, a red-light driven Channelrhodopsin[33]. *H24 > Chrimson* larvae showed a significant preference for the red-light illuminated rectangles within the first minute, while control larvae were randomly distributed (Fig. 3d). Similar to our previous results, the effect was gone after 3 min (Fig. 3d). To test this even further, we challenged larvae to decide between OCT in the darkness and blue light at the opposing side (Supplementary Fig. 5). Here, control larvae are forced to go into the darkness due to their innate light avoidance and innate OCT preference. Control larvae strongly preferred the dark and OCT side over blue light. Experimental larvae, however, showed significantly reduced preference for darkness and OCT (Supplementary Fig. 5). This supports our hypothesis that experimental larvae were challenged to decide between their innate OCT and darkness preference versus an artificially induced internal reward signaling in the blue light. Taken together, optogenetic activation of KCs appears to elicit an internal reward signaling inducing appetitive memory during KC-substitution learning. This is further supported by the change in orientation behavior, reflected by the reduction in turning rate and direction (Supplementary Fig. 3), in larvae with optogenetically activated KCs. Blue light exposure without any gradient may provide an even distributed pleasant environment in experimental larvae due to the optogenetic activation of internal reward signaling. Consequently, experimental larvae reduce turning rate and turning direction, but not speed (Fig. 2a, b, Supplementary Fig. 3)[34].

**DANs of the pPAM cluster mediate internal reward signals.**
Previous studies have shown that dopaminergic cells of the pPAM cluster in larval and adult *Drosophila* mediate internal reward signals during associative conditioning[9,12,13,35]. To test whether internal reward signaling described in our study is based on the activity of pPAM neurons, we repeated the light avoidance

experiment, but this time expressed ChR2-XXL in three out of four pPAM neurons via *R58E02-Gal4* (Fig. 3g)[13]. In line with our previous results, optogenetic activation of pPAM neurons abolished light avoidance in *R58E02 > ChR2-XXL* larvae (Fig. 3f). The effect was even stronger compared with *H24 > ChR2-XXL* larvae (Fig. 3b), as *R58E02 > ChR2-XXL* larvae showed no light avoidance after both 1 and 3 min (Fig. 3f). Further, we tested whether the pPAM-dependent reward signaling is induced by the optogenetic activation of KCs in *H24 > ChR2-XXL* larvae. We used the LexA/LexAop system[36] together with the Gal4/UAS system to activate KCs (*H24 > ChR2-XXL*) and simultaneously ablate pPAM neurons (*R58E02 > reaper*). As expected, optogenetic activation of KCs abolished the innate light avoidance in both genetic controls after 1 and 3 min (Fig. 3i). In contrast, experimental larvae with ablated pPAM neurons showed light avoidance after 3 min, indicating that optogenetic activation of KCs induces internal reward signaling via the activation of pPAM neurons (Fig. 3i). To visualize potential overlaps between KC and pPAM arborizations, we used the GRASP technique which allows to label potential synaptic connections by reconstituted GFP fluorescence which becomes visible when the neurons of interest are in close vicinity[37]. GFP fluorescence was exclusively detectable in the MBs, mostly in the medial lobe (Fig. 3j)[13]. This supports the hypothesis that the optogenetic activation of KCs induces pPAM-dependent reward signaling directly at the level of the MBs, as elsewhere no signal was detectable.

**KC-to-DAN signaling sufficiently induces memory expression.**
Our results suggest that the optogenetic activation of KCs induces reward signaling involving a recurrent pPAM cluster loop. Previous studies already showed that pPAM neurons are important MBINs to establish larval olfactory memories in *Drosophila*[12,13]. Thus, we tested whether the recurrent KC-to-pPAM loop is important to establish the appetitive memory observed in our KC-substitution learning experiment (Fig. 4). First, we tested whether the ablation of pPAM neurons affects appetitive memory established via optogenetic activation of KCs (*H24 > ChR2-XXL;R58E02 > reaper*). Indeed, experimental larvae showed no appetitive memory in contrast to the genetic controls (Fig. 4b). Second, to verify these results, we activated KCs and simultaneously downregulated the dopamine receptor DopR1 in KCs (*dcr2;H24 > ChR2-XXL;DopR1-RNAi*). DopR1 is necessary for appetitive olfactory learning[14,38]. Experimental larvae showed no appetitive memory in contrast to the genetic control (Fig. 4e). Importantly, impaired DA signaling did not affect learning-relevant odor preferences (Fig. 4c, f). Next, we monitored changes of intracellular $Ca^{2+}$ levels in R58E02-positive pPAM neurons upon optogenetic activation of KCs (*H24>ChR2-XXL;R58E02>GCamp6m*). Isolated brains were first imaged with low blue light intensity (~500 µW cm$^{-2}$) to monitor baseline activity; subsequently, brains were exposed to a higher light intensity for 1 min (~1000 µW cm$^{-2}$, Fig. 4g, Supplementary Fig. 6) and fluorescence intensities in pPAM neurons were monitored with low light intensity (~500 µW cm$^{-2}$) after the light pulse. The increase in light intensity was sufficient to induce a significant increase in $Ca^{2+}$ levels within pPAM neurons due to the activation of ChR2-XXL in KCs (Fig. 4h, i, Supplementary Fig. 6), which confirms the functionality of the KC-to-pPAM connection. Taken together, our results suggest that functional recurrent signaling between MB KCs and pPAM neurons exist in the *Drosophila* larva.

**Feedback signaling to pPAM neurons is independent of ACh.**
Our results suggest KCs signal onto pPAM neurons during larval odor-reward learning. This is in line with a study in adult *Drosophila* showing cholinergic feedback signaling from KCs to DANs of the PPL (protocerebral posterior lateral) cluster during

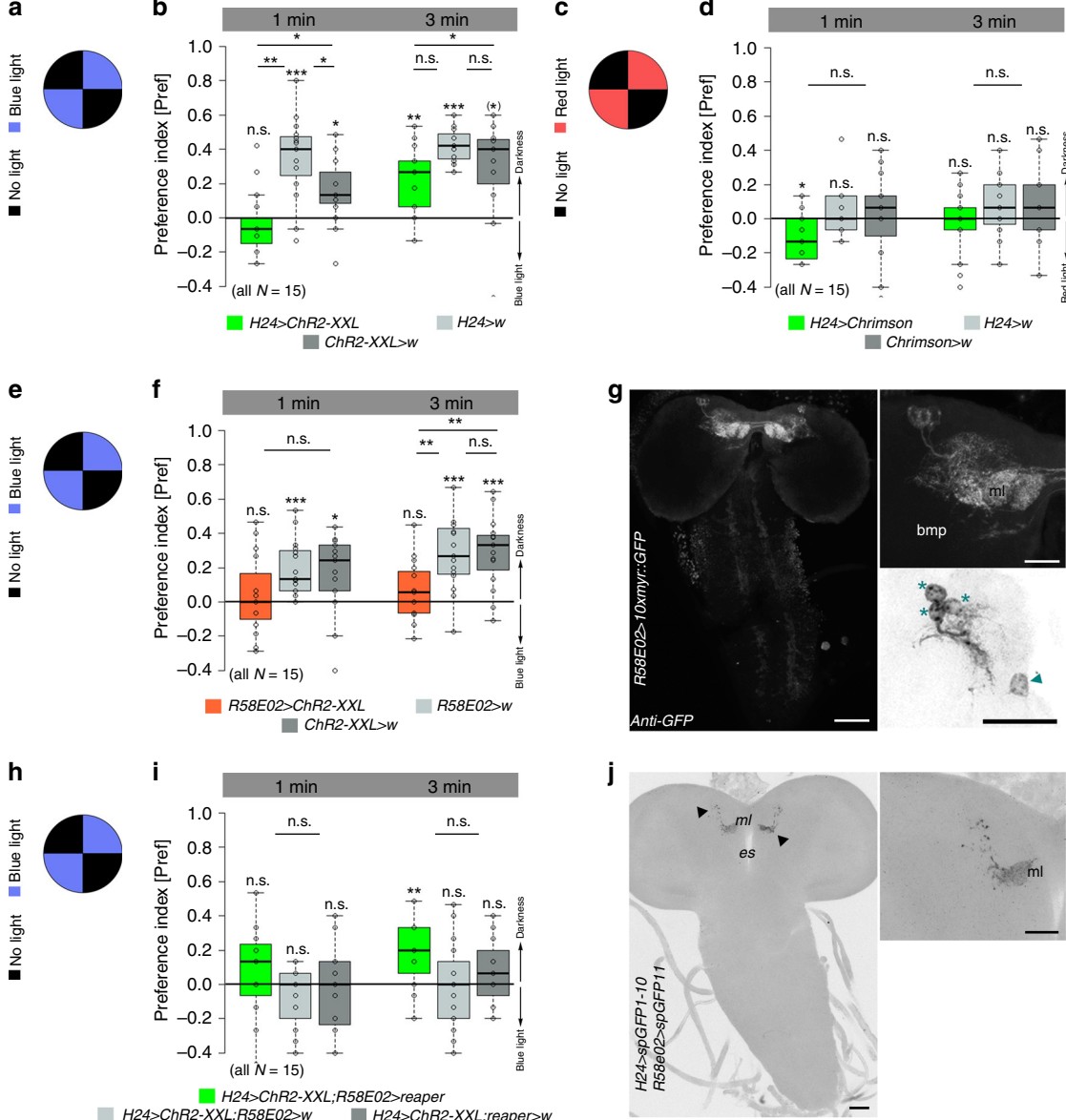

**Fig. 3** Optogenetic activation of KCs induces internal reward signaling. **a, b** Control larvae showed blue light avoidance in a simple choice test. *H24 > ChR2-XXL* larvae do not show any light avoidance after 1 min, but after 3 min suggesting that the blue light-dependent activation of ChR2-XXL induces internal reward signaling. **c, d** In line, *H24 > Chrimson* larvae prefer red light over darkness after 1 min, while genetic controls were randomly distributed. **e, f** Optogenetic activation of pPAM neurons using *R58E02-Gal4* abolished light avoidance. **g** Expression pattern of *R58E02-Gal4* crossed with 10xUAS-*myr:: GFP* and stained with anti-GFP (white). Three DANs of the pPAM (protocerebral anterior medial) cluster (asterisks) and one additional cell body at the midline (arrowhead) are labeled. pPAM neurons innervate the medial lobe of the MBs. **h, i** Optogenetic activation of KCs abolished light avoidance in both genetic controls; in contrast experimental larvae with ablated pPAM neurons showed light avoidance after 3 min, indicating that optogenetic activation of KCs induces internal reward signaling via pPAM neurons. **j** Reconstituted split-GFP between *H24-Gal4* positive neurons and *R58E02-Gal4* positive neurons is only visible at the level of the MBs (arrowheads). bmp: basomedial protocerebrum; ChR2-XXL: channelrhodopsin2-XXL; DANs: dopaminergic neurons; es: esophagus; KCs: Kenyon cells; MBs: mushroom bodies; ml: medial lobe; 10xmyr::GFP: 10xUAS-myristoylated green-fluorescent protein; w: w[1118]. A pairwise Student's *t* test or pairwise Wilcoxon test (both including Bonferroni-Holm correction) was used. Significance levels: n.s.*p* > 0.05, *\*p* < 0.05, \*\**p* < 0.01, \*\*\**p* < 0.001. Scale bars: 50 µm and 25 µm for higher magnifications. Data are mainly presented as box plots, with 50% of the values of a given genotype being located within the box, and whiskers represent the entire set of data. No data were excluded. Outliers are indicated as open circles. The median performance index is indicated as a thick line within the box plot

aversive odor-shock learning[39]. We thus tested whether KC-to-pPAM signaling in the *Drosophila* larva likewise depends on acetylcholine (ACh). For this, we used RNAi lines to specifically knockdown different subunits of the ACh receptor (AChR) in pPAM neurons (*dcr2;R58E02 > AChR-RNAi*[39]) during normal odor-sugar conditioning (Fig. 5). However, knockdown of AChR subunits α1, α4, α5, or α6, respectively, did not affect odor-sugar

learning, suggesting that larval KC-to-pPAM feedback signaling is independent of ACh signaling (Fig. 5).

**Feedback signaling to pPAM neurons is dependent on sNPF.** Immunostainings and transcriptomic studies revealed that MB KCs express short neuropeptide F (sNPF) in addition to ACh[17,18,40]. Therefore, we tested whether KC-to-pPAM signaling

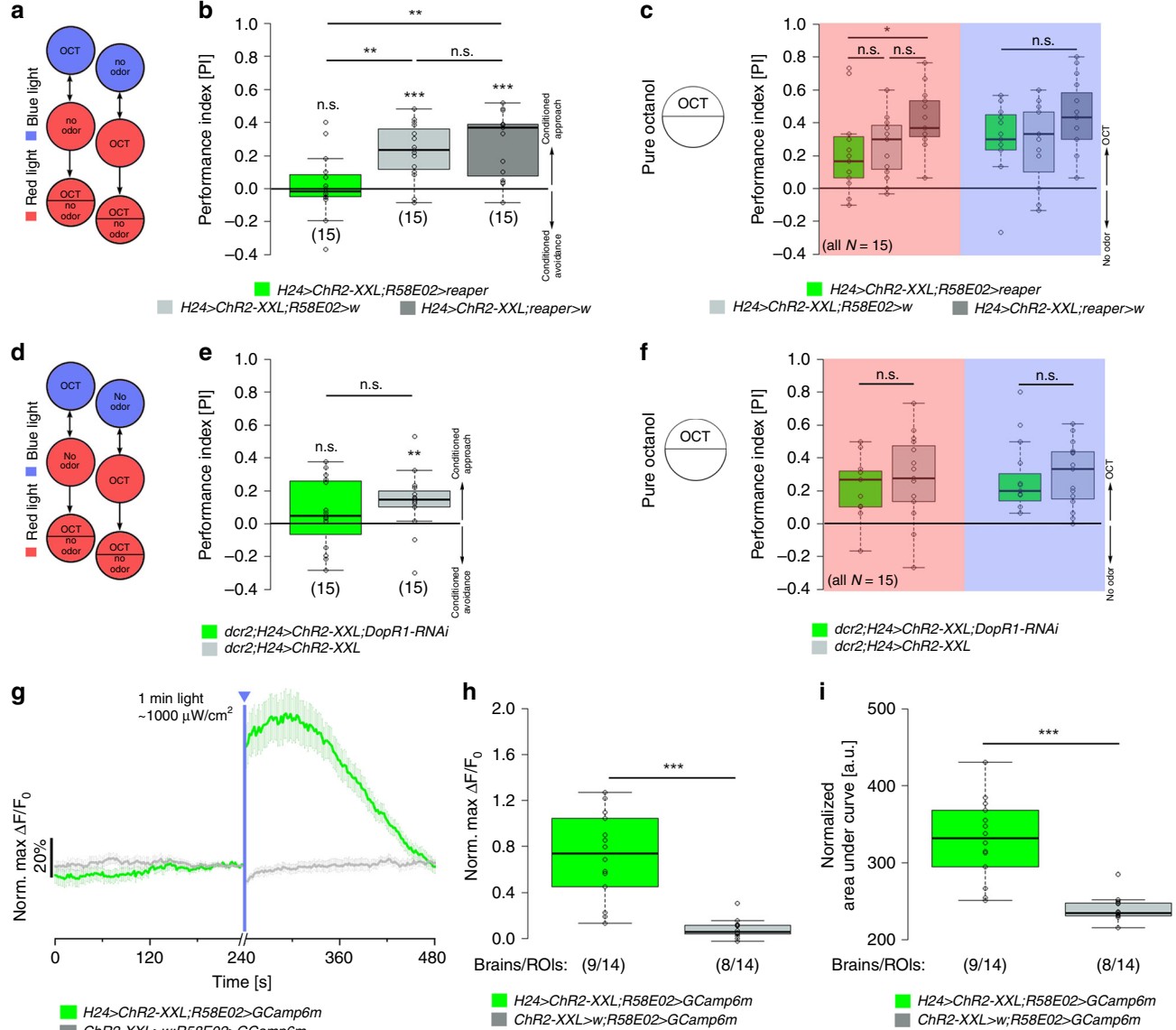

**Fig. 4** Artificially induced memory expression depends on KC-to-pPAM signaling. **a**, **b** Optogenetic activation of KCs and simultaneous ablation of pPAM neurons via the expression of the apoptotic gene reaper impairs optogenetically induced appetitive memory. **c** Innate odor preference is unchanged in experimental larvae under red and blue light. **d**, **e** Similarly, optogenetic activation of KCs and simultaneous knockdown of the dopamine receptor DopR1 impairs appetitive memory formation. **f** Innate odor preference is unchanged in experimental larvae under red light (red rectangle) and blue light (blue rectangle). **g–i** pPAM neurons respond with a significant increase in fluorescence intensity to optogenetic activation of MB KCs (green) both in the maximum values (**h**) and area under the curve (**i**) verifying a functional KC-to-pPAM feedback connection ($N = 14$ ROIs from 9 brains). An increase in fluorescence intensity was absent in genetic controls (gray; $N = 14$ ROIs from 8 brains). ChR2-XXL: channelrhodopsin2-XXL; KCs: Kenyon cells; MB: mushroom body; OCT: octanol; ROI: Region of interest; w: w[1118]. The number below the box plots refers to the $N$ number. A pairwise Student's $t$ test or pairwise Wilcoxon test (both including Bonferroni-Holm correction) was used. Significance levels: n.s. $p > 0.05$, *$p < 0.05$, **$p < 0.01$, ***$p < 0.001$. Error bars (**g**) represent the standard error of the mean. Data are mainly presented as box plots, with 50% of the values of a given genotype being located within the box, and whiskers represent the entire set of data. No data were excluded. Outliers are indicated as open circles. The median performance index is indicated as a thick line within the box plot

is dependent on sNPF via specific knockdown of the sNPF receptor (sNPFR) in pPAM neurons during odor-sugar conditioning (Fig. 6a, b). Indeed, experimental larvae (*R58E02 > dcr2; sNPFR-RNAi*) showed significantly reduced memory scores compared with genetic controls (Fig. 6b). Importantly, specific knockdown of sNPFR in pPAM neurons neither affected innate odor preference (Fig. 6d) nor sugar preference (Fig. 6f). To confirm this connection to be functional, we monitored intracellular Ca²⁺ levels of pPAM neurons and found significantly higher fluorescence intensity levels in response to sNPF compared

with the vehicle control (Fig. 6g–i). Next, we addressed whether sNPF signaling to pPAM neurons potentially originates from MB KCs. We first confirmed sNPF expression in larval KCs by antibody staining (Fig. 6j). Further, electron-microscopic data revealed the presence of dense core vesicles in KCs at the medial lobe next to pPAM neurons (Fig. 6k; source of electron-microscopic volume: ref. [41]) supporting a potential peptidergic KC-to-pPAM signaling. Finally, we challenged the optogenetically induced memory of the KC-substitution experiment (Fig. 6l) by a simultaneous and specific knockdown of *amontillado* in KCs.

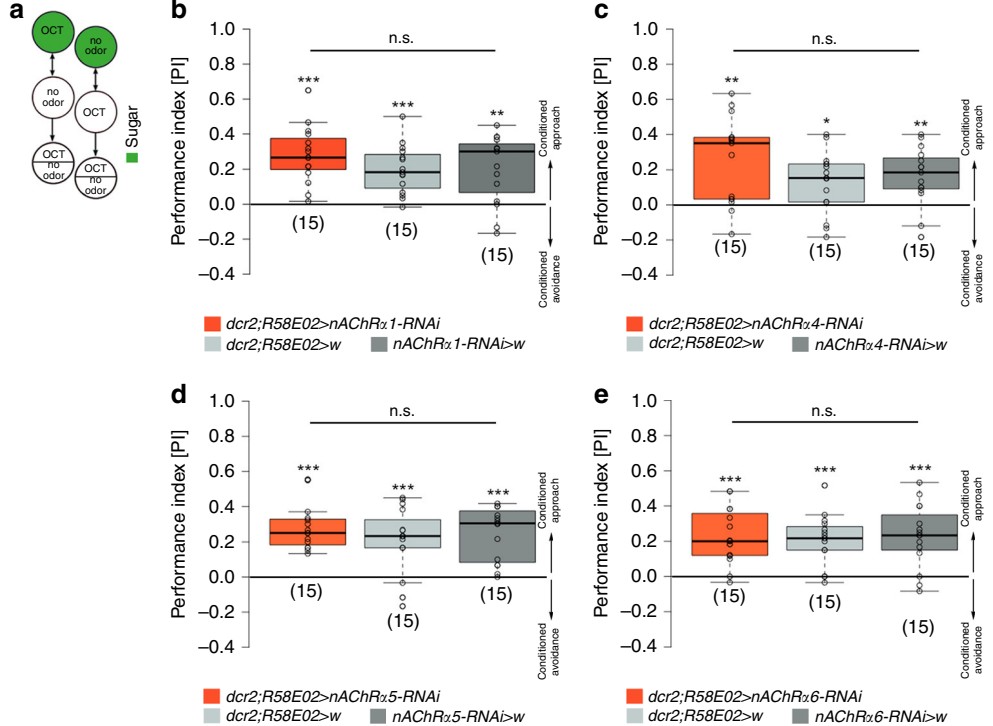

**Fig. 5** KC-to-DAN feedback signaling is independent of ACh. **a** Illustration of the 1-odor reciprocal training regime used in this study (odor-sugar learning). **b**–**e** Specific knockdown of acetylcholine receptor subunits α1 (**b**), α4 (**c**), α5 (**d**), α6 (**e**), respectively, in pPAM neurons does not affect odor-sugar learning. AChR: acetylcholine receptor; dcr2: dicer2; OCT: octanol; w: w[1118]. The number below the box plots refers to the N number. A pairwise Student's t test or pairwise Wilcoxon test (both including Bonferroni-Holm correction) was used. Significance levels: n.s.$p > 0.05$, *$p < 0.05$, **$p < 0.01$, ***$p < 0.001$. Data are mainly presented as box plots, with 50% of the values of a given genotype being located within the box, and whiskers represent the entire set of data. No data were excluded. Outliers are indicated as open circles. The median performance index is indicated as a thick line within the box plot

*Amontillado* encodes the pro-protein convertase dPC2, an enzyme necessary for neuro- and enteroendocrine peptide production[42,43]. Indeed, the lack of peptide processing in KCs (*H24 > ChR2-XXL;amon-RNAi*) resulted in an impairment of memory expression, supporting that the optogenetically induced memory in *H24 > ChR2-XXL* larvae depends on peptidergic signaling (Fig. 6m). Importantly, the knockdown of peptide processing did not affect innate odor preferences in experimental larvae (Fig. 6n). To confirm this finding, we monitored fluorescence intensity in pPAM neurons upon KC activation in combination with *amon-RNAi* to impair peptide processing in the mushroom body. Indeed, the increase in fluorescence intensity in pPAM neurons was significantly reduced due to the lack of peptide expression in KCs (Supplementary Fig. 7). As sNPF is so far the only peptide known to be expressed in KCs, we argue that "peptidergic signaling" is equal to "sNPF signaling".

**Memory stabilization is induced by recurrent MB signaling.** The larval MB connectome revealed a canonical circuit motif in each MB compartment, with suggested MBIN-to-KC and KC-to-MBON synapses, but also unknown recurrent KC-to-MBIN and MBIN-to-MBON synapses[11]. Our data suggest that the KC-to-pPAM connection is functional, which is in line with a recent report in adult *Drosophila*[39]. To test whether KC-to-pPAM feedback signaling affects normal odor-sugar learning, we exposed experimental larvae to a real sugar stimulus and a simultaneous optogenetic activation of KCs via ChR2-XXL ("blue light + sugar"), while control flies were trained with sugar only under red light ("sugar only") or with optogenetic activation of KCs but without sugar ("blue light only"), respectively (Fig. 7a). *H24 > ChR2-XXL* larvae trained to associate OCT with sugar

under red light (normal sugar learning) showed significant appetitive memory expression immediately after training, while memory expression was abolished 15 min after training (Fig. 7b, c). Similarly, significant memory expression in the KC-substitution experiment was abolished 15 min after training ("blue light only"; Fig. 7b, c). *H24 > ChR2-XXL* larvae that received sugar stimulation in combination with artificial KC activation during training showed comparable appetitive memory expression immediately after training. However, the appetitive memory expression was still detectable 45 min after training, suggesting a change in memory persistence based on artificial activation of the MB network (Fig. 7b, c).

## Discussion
Optogenetic KC activation is sufficient to induce olfactory appetitive memory. Based on these findings, we suggest to modify the current scheme of the appetitive olfactory learning circuitry as following: during larval odor-sugar conditioning, CS (odor) information is mediated by projection neurons to the MB calyx, while simultaneously US (sugar) information is mediated via dopaminergic pPAM neurons to the medial lobe. The coincident odor-dependent $Ca^{2+}$ influx and DA release induces synaptic plasticity between KCs and MBONs[4]. Importantly, cAMP-dependent plasticity in KCs increases relative KC responses to the sugar-paired odor specifically for appetitive conditioning as shown for adult *Drosophila*[44]. Such a mechanism may explain why optogenetic activation of KCs specifically induces appetitive memory formation in our study.

Our new data suggest, that in addition pPAM neurons receive input from KCs during training. Optogenetic activation of KCs was sufficient to activate the dopaminergic pPAM neurons

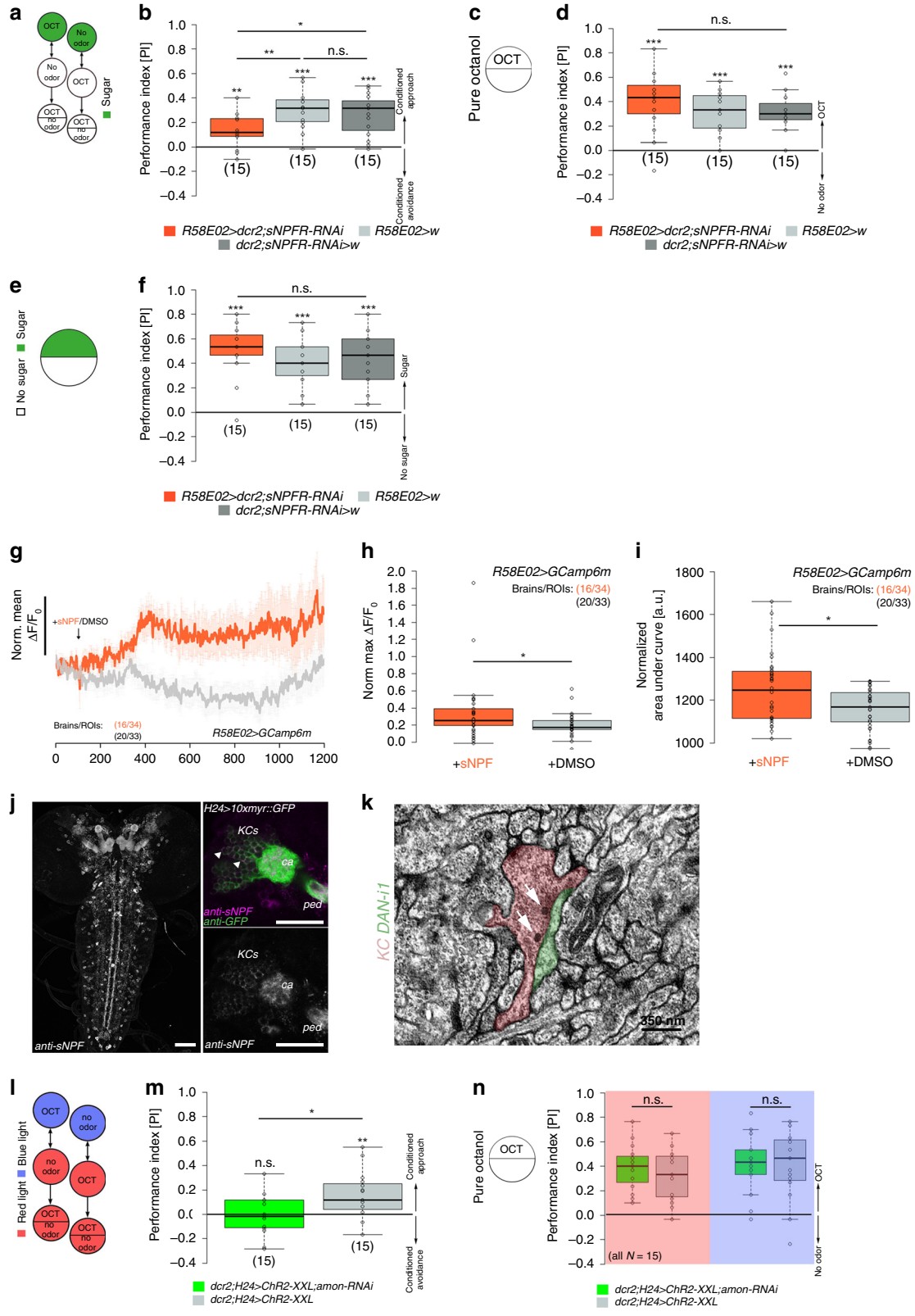

indicating a functional connection of KCs-to-pPAM, which was anatomically identified in the larval MB connectome[11]. Yet, during odor-sugar training a recurrent circuit of KCs and pPAMs could become active, instruct and consolidate an appetitive olfactory memory. Indeed, when this loop was blocked, either through DopR1 knockdown in KCs or ablation of pPAM neurons, the artificial induction of appetitive memory was impaired.

Several studies on the adult fly convincingly showed that neurotransmission from KCs is necessary not only during memory retrieval, but also during training. The current assumption is that memory formation occurs in αβ neurons, while the memory is stabilized by the recurrent activity in α′β′ neurons, DPM (dorsal paired medial) neurons, and αβ neurons during and after training[45,46]. This model was further supported

**Fig. 6** KC-to-DAN feedback signaling depends on sNPF. **a** Illustration of the 1-odor reciprocal training regime used in this study (odor-sugar learning). **b–f** Knockdown of the sNPF receptor in pPAM neurons significantly impairs odor-sugar conditioning (**b**), while innate odor preference (**d**) and sugar preference (**f**) is not affected. **g–i** pPAM neurons respond with a significant increase in intracellular fluorescence intensity to sNPF application (orange: sNPF application, $N = 34$ ROIs from 16 brains; gray: DMSO application, $N = 33$ ROIs from 20 brains) both in the maximum values (**h**) and area under the curve (**i**). **j** sNPF is expressed throughout the nervous system, but particularly in KCs (arrowheads). **k** Electron-microscopic data showing dense core vesicles (white arrows) in a KC (red) at the axonic region of the medial lobe next to the DAN-i1 pPAM neuron (green). **l–n** Knockdown of the protein convertase 2 encoded by amontillado impairs the optogenetically induced memory (**m**) in our KC-substitution assay (**l**), while odor preferences are not affected (**n**). amon: amontillado; ca: calyx; ChR2-XXL: channelrhodopsin2-XXL; dcr2: dicer2; DMSO: dimethyl sulfoxide; KCs: Kenyon cells; 10xmyr:GFP: 10xUAS-myristoylated green-fluorescent protein; OCT: octanol; ped: peduncle; ROI: region of interest; sNPF: short neuropeptide F; w: w[1118]. The number below the box plots refers to the $N$ number. A pairwise Student's $t$ test or pairwise Wilcoxon test (both including Bonferroni-Holm correction) was used. Significance levels: [n.s.]$p > 0.05$, *$p < 0.05$, **$p < 0.01$, ***$p < 0.001$. Error bars (**g**) represent the standard error of the mean. Data are mainly presented as box plots, with 50% of the values of a given genotype being located within the box, and whiskers represent the entire set of data. No data were excluded. Outliers are indicated as open circles. The median performance index is indicated as a thick line within the box plot

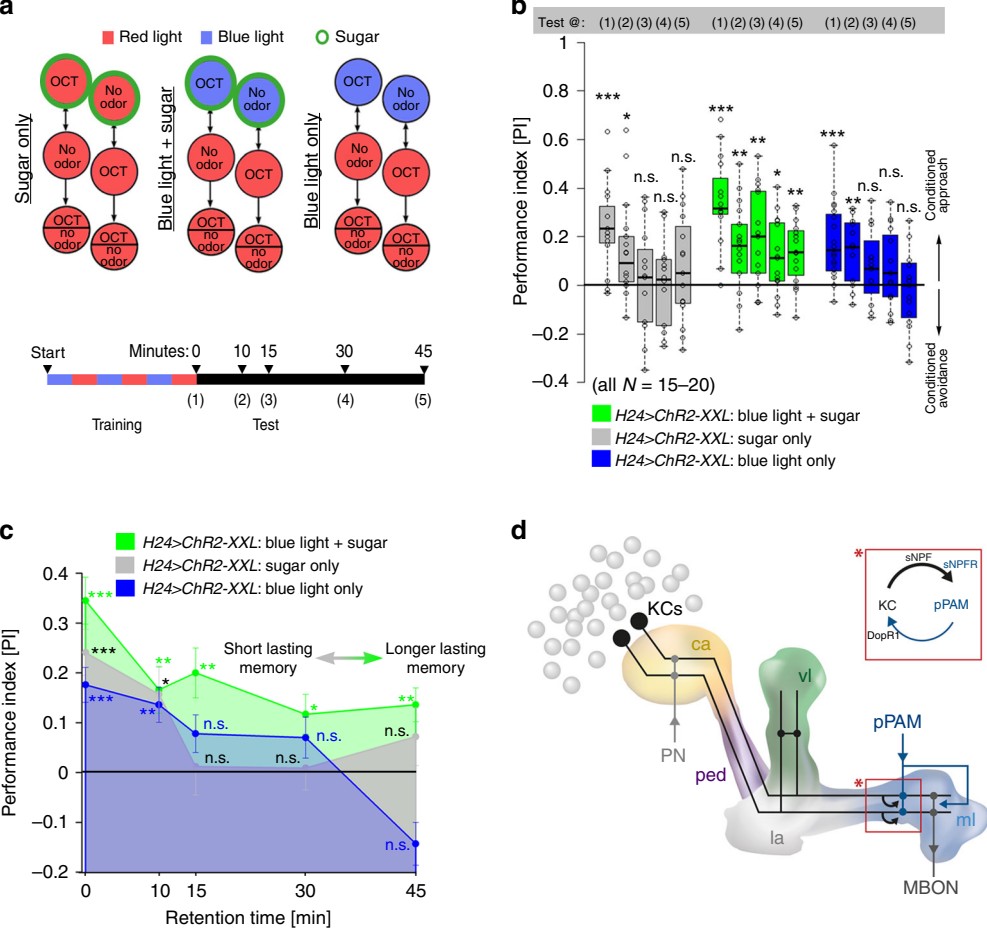

**Fig. 7** KC activation during training increases memory stabilization. **a** Schematic drawing of the training and test regime: *H24 > ChR2-XXL* larvae were either trained for odor-sugar associations under red light, in the KC-substitution experiment, or for odor-sugar associations with paired artificial KC activation (green circle: sugar exposure). Appetitive memory was either tested immediately after training, or 10, 15, 30, and 45 min after training. **b**, **c** *H24 > ChR2-XXL* larvae trained under red light ("sugar only"; gray) showed significant performance scores immediately after training as well as 10 min after training. No memory was detectable 15 min or later after training. Similarly, memory scores were abolished 15 min after training in the KC-substitution experiment ("blue light only"). *H24 > ChR2-XXL* larvae trained with blue light exposure and sugar stimulus during training ("blue light + sugar") showed significant memory 45 min after training. **d** Schematic of the mushroom body circuit focusing on the connectivity with dopaminergic input neurons. Recurrent signaling between KCs and dopaminergic pPAM neurons exists via the sNPF receptor and dDA1 receptor (DopR1). ChR2-XXL: channelrhodopsin2-XXL; ca: calyx; DopR1: dopamine receptor 1; KCs: Kenyon cells; la: lateral appendix; MBON: mushroom body output neuron; ml: medial lobe; OCT: octanol; ped: peduncle; PN: projection neuron; sNPF: short neuropeptide F; vl: vertical lobe. A pairwise Student's $t$ test or pairwise Wilcoxon test (both including Bonferroni-Holm correction) was used. Significance levels: [n.s.]$p > 0.05$, *$p < 0.05$, **$p < 0.01$, ***$p < 0.001$. Error bars (**c**) represent the standard error of the mean. Data are mainly presented as box plots, with 50% of the values of a given genotype being located within the box, and whiskers represent the entire set of data. No data were excluded. Outliers are indicated as open circles. The median performance index is indicated as a thick line within the box plot

by findings that ongoing activity and feedback signaling is necessary for olfactory learning and memory as impaired cholinergic input from KCs to dopaminergic neurons of the PPL cluster reduces aversive odor-electric shock memories in adult *Drosophila*[39]. In this sense, KC-to-DAN feedback signaling may affect neuronal activity in MBINs and by that enhances the release of DA to the MBs beyond the level induced by the US stimulus alone[39]. Likewise, dopaminergic MBINs receive excitatory input from MBONs during courtship training and training for odor-sugar associations, which in turn leads to a persistent activity of these DANs and an increase in DA release onto the MBs. Thus, MBONs are not only necessary for memory retrieval, but also for memory acquisition and consolidation. Consequently, a longer-lasting memory is formed by the modification of KC-to-MBON signaling through the recurrent MBON-to-DAN feedback loop[15,47]. Taken together, within the MB circuitry direct recurrent KC-to-DAN loops together with one step recurrent KC-MBON-DAN loops exist to shift an initial nascent memory into a stable memory.

Our data now extend the model of recurrent signaling in the MB circuitry and suggest that different transmitter or modulator systems can be used. KC-to-PPL feedback signaling during odor-shock learning depends on ACh in adult *Drosophila*[39]. We show that KC-to-pPAM signaling in the larva depends on sNPF. Interestingly, also in adults sNPF is expressed in a high number of KCs, and acts as a neuromodulator for odor memories[17]. Similar to our results, knockdown of sNPF in KCs impaired adult odor-sugar memories. Interestingly, while a pan-neuronal knockdown of sNPFR impaired odor-sugar memories, a specific knockdown of the receptor in KCs had no effect, indicating that sNPF release from KCs is not part of an intrinsic KC-to-KC signaling[17]. One may assume that KC-to-pPAM signaling via sNPF is conserved beyond *Drosophila* development and provides a mechanism to modulate appetitive memory acquisition and stabilization. However, the available data do not exclude additional signaling via KC-to-MBON-to-pPAM circuits.

Does recurrent signaling within the larval mushroom bodies allow the formation of different types of memories? Our results indicate that a paired exposure to sugar and optogenetic activation of KCs increases memory persistence. Larvae trained for only odor-sugar associations under red light showed memory expression up to 10 min, while 15 min after training memory expression was abolished. On the contrary, artificial activation of KCs combined with sugar exposure during training was sufficient to induce a longer-lasting appetitive olfactory memory. Memory expression was still detectable 45 min after training. Thus, it is well possible that recurrent signaling in the MB circuitry, including KC-to-pPAM activation during training, supports the formation of longer-lasting memories beyond developmental stages.

Following this reasoning dopaminergic pPAM neurons may not only become activated by the US during training, but also via the KC-to-pPAM signaling. In consequence, different activity levels in pPAM neurons based on situation-dependent accumulated input signals adjust intracellular cAMP levels, either triggering a short-lasting memory (by a cAMP-dependent transient increase in protein kinase A (PKA)) or longer-lasting memories (by more stable cAMP-dependent elevation of PKA)[48,49]. Internal information may thus be appropriate to regulate the type of memory which is formed. In line, under food restriction adult *Drosophila* preferentially show anesthesia-resistant memory rather than long-term memory, as this type of memory is independent of energy-demanding protein biosynthesis and therefore at low cost[50,51]. Further, in both adult and larval *Drosophila*, neurons expressing dNPF (*Drosophila* neuropeptide F; an orthologue of mammalian

NPY) modulate memory processes to match information about the current reward stimulus with the internal state of the animal. A conditional activation of dNPF neurons during training reduced the acquisition of odor-sugar memories in the *Drosophila* larva[52]. In contrast, odor-sugar learning in flies is most efficient when flies are food deprived. dNPF neurons modulate the expression of odor-sugar memories via feedforward inhibition through dopaminergic MBINs and the MBON circuitry. In hungry flies, specific MBONs (MV2) are inhibited by the action of dNPF neurons through MP1 DANs, which elicits the expression of odor-sugar memories and thus conditional approach behavior[53]. Consequently, an intrinsic feedback signaling via the neuromodulator sNPF in addition to extrinsic input via dNPF neurons to the dopaminergic MBINs allows a dynamic intrinsic and extrinsic modulatory system to different compartments of the MBs. Depending on experience and the internal state, the same sensory input (e.g., a sugar stimulus) may drive a different conditional behavioral output (either approach, avoidance, or indifference) and different temporal forms of associative olfactory memories (short-term memory, anesthesia-resistant memory, or long-term memory)[54,55].

Our assumption is now that recurrent signaling within the larval MB circuit allows the larva to constantly adapt appetitive memory formation and retrieval to the current internal state and external situation, mainly based on the modulation of pPAM neurons, and by that to incorporate whether the expected gain is sufficient to store and recall a long-lasting associative memory. In humans, DA was shown to modulate a post-encoding consolidation process with respect to episodic memory persistence[56]. The molecular consolidation process is based on DA-dependent protein-synthesis, which elicits long-term plasticity in the hippocampus. Similarly, in rodents the activation of different DA receptors in the hippocampus is critical at or around the time of memory encoding to modulate memory persistence[57]. Thus, activation of recurrent signaling routes within a neuronal memory circuit via DANs appears as a conserved mechanism to consolidate memories in insects and vertebrates.

## Methods

**Fly strains**. Flies were cultured according to standard methods. In short, vials were kept under constant conditions with 25 °C and 60% humidity in a 12:12 light:dark cycle. Driver lines used in this study were H24-Gal4 (chromosome III), OK107-Gal4 (IV), 201y-Gal4 (II), MB247-Gal4 (III), R58E02-Gal4 (III), and R58E0-LexA (II). UAS lines included in this study were 10xUAS-IVS-myr::GFP (III), UAS-mCD8::GFP (II), UAS-ChR2-XXL (II), UAS-AChRα1-RNAi (III),UAS-AChRα1-RNAi (III), UAS-AChRα5-RNAi (III), UAS-AChRα6-RNAi (III), UAS-amon-RNAi78b (III), UAS-amon-RNAi28b (II), UAS-sNPFR-RNAi (III), 20xUAS-CsChrimson (I), UAS-DopR1-RNAi (III), LexAop-reaper (III), and 13xLexAop-IVS-GCamp6m,20xUAS-CsChrimson (III). Genetic controls were obtained by crossing Gal4-driver/Lex-driver and UAS-effector/LexAop-effector lines to w1118.

In case for optogenetic experiments, fly strains were covered with aluminum foil prior to the experiment to prevent a constant activation of neurons expressing Channelrhodopsin2. Further, we omitted to add retinal to our food as Channelrhodospin2-XXL was shown to be efficient without addition of retinal[23]. However, in experiments using UAS-Chrimson, we added (~200 mM) all-*trans* retinal to food[23,30,58].

**Associative conditioning**. Appetitive olfactory learning was tested using standardized assays[5,59]. Learning experiments were performed on plates filled by a thin layer of pure 1.5% agarose solution (Roth, 9012-36-6). We mainly used a one-odor reciprocal training design[20], where larvae were exposed to 10 μl of 1-octanol (OCT, Sigma, 111-87-5). We also performed one experiment using the two-odor reciprocal training design, where 10 μl of amylacetate diluted 1:40 in paraffin oil (AM, Merck, 628-63-7) was exposed to the larvae opposing OCT. Odorants were loaded into custom-made Teflon containers (4.5 mm diameter) with perforated lids.

For KC-substitution learning experiments, a first group of 30 larvae was trained to associate OCT to blue light exposure (OCT+). After 5 min the larvae were transferred to a second Petri dish containing no odor under red light (NO). Simultaneously, a second group of larvae was trained reciprocally with blue light exposure coupled to no odor (NO+/OCT). After three training cycles, larvae were immediately transferred to the test plate, where OCT was presented on one side of

the dish. To test the persistence of the established memory, larvae were tested 10, 15, 30, or 45 min after the training. After 3 min, larvae were counted in the OCT side (#OCT), the no odor side (#NO), and in the neutral zone (for further details, a video is provided in ref. [59]). The procedure was similar for the two-odor reciprocal training design (OCT+/AM and OCT/AM+). The preference index was calculated by subtracting the number of larvae on the OCT side from the number of larvae on the no odor side (or AM side), divided by the total number of animals (including the larvae counted in the neutral zone).

For odor-sugar learning or sugar test plates, 2 M fructose (Roth, 57-48-7) was added to the agarose solution. To obtain test plates containing amino acids, we added aspartic acid (Roth, 56-84-8) to the agarose solution.

$$PREF_{OCT+/NO} = (\#OCT - \#NO)/\#TOTAL \quad (1)$$

$$PREF_{OCT/NO+} = (\#OCT - \#NO)/\#TOTAL \quad (2)$$

Subsequently, we calculated the performance index (PI). Positive PIs indicate appetitive learning, while negative PIs indicate aversive learning.

$$PI = \left(PREF_{OCT+/NO} - PREF_{OCT/NO+}\right)/2 \quad (3)$$

For optogenetic manipulation in all behavioral experiments (using UAS-ChR2-XXL) we used 475 nm light-emitting diodes (LED) with a light intensity of ~1300 μW cm$^{-2}$, or red light with 620 nm and the intensity of 50 μW cm$^{-2}$ for UAS-Chrimson. To induce optogenetic activation, the light-emitting diodes were placed ~45 cm above the Petri dish, while all other steps of the experimental procedure were done under red light (or complete darkness for UAS-Chrimson).

**Olfactory preference tests**. To test larvae for their innate odor response, an odor container was placed on one side of the Petri dish containing 1.5% agarose to induce a choice test. In line with the learning experiment, larvae were counted on each side after 3 min. For optogenetic manipulation the Petri dish was placed in blue light, for control experiments the choice test was done under red light. The preference index was calculated by subtracting the number of larvae on the odor side (#Odor) from the number of larvae on the no odor side (#NO), divided by the total number of animals (including the larvae counted in the neutral zone). In each test, we used a group of 30 larvae.

$$PREF_{Odor}/NO = (\#Odor - \#NO)/\#TOTAL \quad (4)$$

Negative PREF values indicate an avoidance of the odor, whereas positive PREF values represent an attractive response.

**Gustatory preference tests**. To test larvae for their innate gustatory response during optogenetic activation, one-half of the Petri dish was filled with 1.5% pure agarose, while the other half was filled with 1.5% agarose containing 2 M fructose (Roth, 57-48-7). In line with the learning experiment, larvae were counted on each side after 3 min. For optogenetic manipulation the Petri dish was placed in blue light, for control experiments the choice test was done under red light. The preference index was calculated by subtracting the number of larvae on the sugar side (#Sugar) from the number of larvae on the no sugar side (#NS), divided by the total number of animals (including the larvae counted in the neutral zone). In each test, we used a group of 30 larvae.

$$PREF_{Sugar}/NS = (\#Sugar - \#NO)/\#TOTAL \quad (5)$$

Negative PREF values indicate sugar avoidance, whereas positive PREF values represent an attractive response.

**Light preference tests**. To test larvae for their response to blue or red light, respectively (and by that to optogenetic manipulation), the Petri dish containing 1.5% agarose was covered by a lid, divided into two transparent and two shaded quarters, respectively. The preference index was calculated by subtracting the number of larvae on the dark side (#DS) from the number of larvae on the illuminated side (#Light), divided by the total number of animals (including the larvae counted in the neutral zone). In each test, we used a group of 30 larvae.

$$PREF_{DS} = (\#DS - \#Light)/\#TOTAL \quad (6)$$

Positive PREF values indicate light avoidance, whereas negative PREF values represent approach toward the illuminated side.

**Locomotion assay**. For the locomotion assay we used the FIM (FTIR-based Imaging Method) tracking system[27]. Recordings were made by a monochrome industrial camera (DMK27BUP031) with a Pentax C2514-M objective in combination with a Schneider infrared pass filter, and the IC capture software (www.theimagingsource.com). To analyze larval locomotion, a group of 10 larvae was recorded on 1.5% agarose for 2 min. During the first minute, larvae were exposed to red light, while they were exposed to blue light for the second minute for optogenetic activation of KCs. We analyzed the following parameters using the FIM track software: accumulated distance, velocity, number of stops, and number of bendings.

**Immunofluorescence**. Immunofluorescence studies were performed as follows[60]: 5–6 day old larvae were dissected in phosphate buffer saline (PBS) or HL3.1 (pH 7.2)[61], fixated in 4% paraformaldehyde in PBS for 40 min, washed in PBS with 0.3% Triton-X 100 (PBT), and afterwards blocked in 5% normal goat serum in PBT. Specimens were incubated in primary antibody solution containing polyclonal rabbit anti-GFP antibody (A6455, Molecular Probes, dilution 1:1000) or monoclonal mouse anti-GFP (A11120, Molecular Probes, dilution 1:250), polyclonal rabbit anti-sNPFp (ref. [40], dilution 1:1000) and 3% normal goat serum in PBT for one to two nights at 4 °C. Then brains were washed six times in PBT and incubated for one night at 4 °C in secondary antibody solution containing goat anti-rabbit Alexa 488 (Molecular Probes, dilution 1:250) or combined goat anti-mouse DyLight 488 (Jackson ImmunoResearch, dilution 1:250) and goat anti-rabbit Alexa 635 (Molecular Probes, dilution 1:250). Finally, specimens were rinsed six times in PBT and mounted in 80% glycerol or Vectashield mounting medium (Vector Laboratories, USA). Until scanning with a Leica SP8 confocal light scanning microscope, brains were stored in darkness at 4 °C. Image processing was performed with Fiji[62] and Adobe Photoshop CS6 (Adobe Systems, USA).

**Functional imaging**. To monitor intracellular Ca$^{2+}$ levels in dopaminergic neurons of the pPAM cluster in response to the optogenetic activation of MB KCs, LexAop-GCaMP6m was expressed via R58E02-LexA and UAS-ChR2-XXL via H24-Gal4. Two to four larval brains were dissected and subsequently put in a Petri dish containing 500 μl hemolymph-like HL3.1 saline solution. Before the experiment, brains were maintained for at least 30 min in complete darkness for settling. Specimens were imaged with an AXIO Examiner D1 upright microscope (Carl Zeiss AG, Germany) with a Zeiss W Plan-Apochromat x20/1.0 water immersion objective and a SPECTRA-4 hybrid solid state LED source (Lumencor, USA). Images were taken with a PCO.edge 4.2 m sCMOS camera (PCO AG, Germany) at a frame rate of 0.5 Hz using a Chroma ET-GFP emission filter, and analyzed by VisiView 3.0.

Brains were excited with 475 nm light with an intensity of around 500 μW cm$^{-2}$ and an exposure time of 80 ms at 2× binning to monitor fluorescence intensity for 1202 s (601 time points). After 300 s (150 time points), imaging was paused, and specimens were exposed to a 1 min continuous light pulse of 1000 μW cm$^{-2}$ (475 nm) for optogenetic activation of MB KCs. Afterwards monitoring of fluorescence intensity in pPAM neurons was continued at lower light intensity (500 μW cm$^{-2}$, 475 nm, time points 151–480), and paused again to give another 1 min continuous light pulse (1000 μW cm$^{-2}$, 475 nm, time point 480). Finally, monitoring was continued for at least 240 s with lower light intensity (500 μW cm$^{-2}$, 475 nm, time points 481–601; see Supplementary Fig. 6).

The same light regime was used for the Chrimson activation of KCs (Supplementary Fig. 7). To analyze fluorescence intensity changes in pPAM neurons, regions of interest were drawn over the cell bodies (Supplementary Fig. 6) and changes in fluorescence intensity were calculated after background subtraction:

$$\Delta F/F_0 = (F_n - F_0)/F_0 \quad (7)$$

$F_n$ = fluorescence at time point $n$, $F_0$ = baseline fluorescence value calculated from time points 475–479 (950–958 s) before the second light pulse. Mean maximal $\Delta F/F_0$ and the area under the curve after normalization were calculated for both groups.

For analysis we only used imaging data of time points 360–601 (720–1202 s; Fig. 4g–i, Supplementary Fig. 6) as we do see a response to the first light (time points 1–150) after resting in complete darkness (Supplementary Fig. 6).

For Ca$^{2+}$ imaging and sNPF peptide application, brains of 5–6 days old R58E02 > GCamp6m larvae were dissected and mounted in 405 μl HL3.1 Ringer solution. sNPF (sNPF-1: AQRSPSLRLRFamide; Iris Biotech GmbH; Germany) was dissolved to a concentration of 10$^{-5}$ M in HL3.1 containing 0.1% DMSO (dimethyl sulfoxide) and applied manually onto dissected brains during acquisition (final concentration 10$^{-4}$ M). Likewise, we applied HL3.1 containing 0.1% DMSO as vehicle control. To analyze differences between corresponding groups in our imaging experiments, we calculated the maximum values ($\Delta F/F_0$) and the area under the curve after normalization.

**Electron microscopy**. Anatomical data showing dense core vesicles in MB KCs were taken from a single, complete central nervous system of a 6h-old [iso] Canton S G1×w [iso] 5905 larva acquired with serial section transmission electron microscopy at a resolution of 3.8 nm × 3.8 nm × 50 nm, which was first published in ref. [41] with the detailed sample preparation protocol. The volume and the reconstructed neurons can be accessed via the following link: https://l1em.catmaid.virtualflybrain.org/.

**Statistical methods**. Data was analyzed for normal distribution using the Shapiro–Wilk Normality test. To test against chance level, a t-test was used for normally distributed data, a Wilcoxon Signed Rank test for not normally distributed data. For the comparison between genotypes, a pairwise t-test was used for normally distributed data, a pairwise Wilcoxon Rank Sum test was used for not normally distributed data. Pairwise tests included the Bonferroni-Holm correction and were two-sided. All statistical analyses were done with R Studio version 0.99.896 (www.r-project.org). Sample sizes were based on previous reports on

chemosensory learning and memory with moderate to weak effect sizes[3,30,52]. Data plots were done with OriginPro 2016G, b9.3.226 (www.originlab.com) or R Studio version 0.99.896. Data are mainly presented as box plots, with 50% of the values of a given genotype being located within the box, and whiskers represent the entire set of data. No data were excluded. Outliers are indicated as open circles. The median performance index is indicated as a thick line within the box plot. For the persistence of memory as well as imaging traces, data are presented as a line chart, with mean values and the standard error of the mean. Significance levels between genotypes shown in the figures refer to the raw $p$-values obtained in the statistical tests. $N$ numbers (number of groups of larvae) or $n$ numbers (number of individual larvae) are always indicated in the figures, mostly below the plots (in brackets). In none of the experiments, individual larvae or groups of larvae were tested repeatedly. All experiments were done in parallel for the respective experimental group and the appropriate genetic controls. Experimenters were -at least- partially blind to genotypes.

**Reporting summary**. Further information on research design is available in the Nature Research Reporting Summary linked to this article.

## Data availability

All data are available from the authors on request. Connectomic analyses are based on data available with ref. [11].

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

## Acknowledgements

We thank Konrad Öchsner, for technical assistance, Wolf Huetteroth, Katharina Eichler, Björn Brembs, Thomas Raabe, Tim Humberg, and Martin Strube-Bloss for fruitful discussions and/or comments on the paper. We thank Albert Cardona for providing access to the complete electron microscopical volume of the larval brain. We thank Robert Kittel, Georg Nagel, Katharina Eichler, Vivek Jayaraman, the Vienna *Drosophila* resource center, and the Bloomington Stock center for providing flies. This work was supported by the PostDoc Plus fellowship (to D.P.) and a PhD fellowship (to R.L.) from the German Excellence Initiative to the Graduate School of Life Sciences, University of Würzburg, and by the Deutsche Forschungsgemeinschaft (TH1584/1-3 to A.S.T., INST 93/824-1 LAGG to C.W., and DP1979/2-1 to D.P.), and a SCIENTIA fellowship "Bayerische Gleichstellungsförderung: Programm zur Realisierung der Chancengleichheit für Frauen in Forschung und Lehre" (to M.S.), intramural funds by the University of Würzburg (to C.W.), and the University of Würzburg in the funding program Open Access Publishing.

## Author contributions

R.L., M.S., A.S.T., and D.P. conceived and designed the experiments. R.L., M.S., M.P., F.F., A.R., J.H., D.S., and D.P. performed the experiments. R.L., M.S., C.W., and D.P. analyzed the results. R.L., C.W., A.S.T., and D.P. wrote the paper.

## Additional information

**Competing interests:** The authors declare no competing interests.

