## [Peer Review File · Nature Communications]

Reviewers' Comments:

Reviewer #1:

Remarks to the Author:

The main claim of this work is that artificial and massive activation of mushroom body neurons induces appetitive memory through activating pPAM dopaminergic neurons. The finding is somewhat surprising, since small ensembles of these neurons have been well known to represent odors, and therefore the effect of massive activation could have disrupted odor specificity. The manuscript however suffers from the lack of evidence to prove that this recurrent circuit functions under physiological conditions.

1. All the data are based on massive activation of most if not all Kenyon cells. To the best of my knowledge, there is no evidence that this would happen under a natural condition. The authors must show supporting experimental evidence that the proposed circuit is functional during real learning. The authors could transiently block Kenyon cells during conditioning, downregulate acetylcholine receptors in the dopaminergic neurons, and/or monitoring sugar responses of Kenyon cells or the dopaminergic neurons upon Kenyon cell blockade.
2. The authors should tone down the interpretation of figure 5. Currently, there is no evidence showing the "sustained" activity of the recurrent mushroom body circuit.
3. Similarly, the necessity of the recurrent circuit in memory stabilization is unclear.
4. The authors should explain how odor identity is encoded under optogenetic activation of all Kenyon cells (Figure 2C). Is odor discrimination possible also with the other GAL4 driver (OK107)?
5. How can figure 3C suggest that "optogenetic activation of KCs induces pPAM-dependent reward signaling (L270)"?

Reviewer #2:

Remarks to the Author:

This manuscript by Lyutova et. al. tests an idea that a functional recurrent loop exists between Mushroom body Kenyon cells (KCs) and a cluster of dopaminergic neurons (pPAM) in *Drosophila* larva brain. The authors first describe a novel phenomenon that artificial activation of KCs paired with odor cue is sufficient to trigger the appetitive memory, and this phenomenon is pPAM neurons dependent. Then, by using behavioral tests and functional imaging, they test if pPAM dopamine neurons mediate reward information to, and receive feedback signals from KCs. Last, the authors show optogenetic activation of KCs increases the persistence of odor-sugar memory. Based on those observations, they hypothesize that the function of the KC-to-pPAM recurrent loop and the sustained activity in this circuitry is to consolidate memories.

Overall, those observations are of interest and provide new insights for understanding the MB-DAN-MBON circuitry in *Drosophila* larva, but some conclusions still cannot be sufficiently supported by the current evidence in the manuscript.

Major concerns:

- In experiments for Fig. 1, two MB Gal4 lines are used, H24 and OK107, to confirm the involvement of Mushroom Body (MB). However, I would suggest them to further exclude the function of MB by combining H24 or OK107 with MB-Gal80.
- As they said, it is actually 'striking' to see the artificial activation of KCs could replace the sugar

reward to induce an odor-specific appetitive memory, given that the odor identity of CS+ represented in the MB axons would be disrupted by the pan-MB activation. I suggest the authors to provide more results for backing up this key and unusual observation. For example, train the flies with one or two more odors to show the generality of this paradigm. They can also train the flies with one odor but test with a different odor to see if the training process cause a non-specific effect on odor-induced behavior. At very least, this odor identity issue should be well discussed in the discussion section.

- Although the authors claim the KC-substitution learning is sufficient to induce appetitive memory by using the two-odor reciprocal paradigm, but the memory score in fig. 2C is lower than one-odor paradigm and is not significantly different from one of its controls. I wonder their thoughts about the possible reasons?
- For fig 4B', in the text, the authors claim experimental larvae showed no appetitive memory in contrast to the genetic control (page 11: 315), but the two groups seem no significant difference in the figure. Why is there both n.s. and ** signs for the two groups?
- For the functional imaging with optogenetic activation experiments, since the baseline brightness of GCaMP is proportional to illumination power, I was confused why there is no change in the dF/F trace of the control group when the illumination power shift from low to high? Any normalization method used here? Also, for fig. 4C', the authors should compare the data within group but at different illumination level instead of comparing the two groups. The result for Chrimson activation is also confusing. The authors should use a low-level blue light for GCaMP imaging at the beginning of each trial and then add a red light to excite the chrimson leaving the GCaMP unaffected.
- For fig 5, the authors wanted to test the proposed KC-to-pPAM feedback loop affects normal appetitive olfactory learning, and propose its function in changing memory persistency. However, the evidence is not enough to support their assumptions, especially most evidence is based on artificial activation. It is possible that the memory created by artificial pan-MB activation is merely different in nature (stronger maybe) to the real sugar memory, so the observed memory curve can be explained by the co-existence of a strong and a weak memory traces created by artificial activation and real odor-sugar association. It will be more convincing to show some blocking experiments to unveil the function of MB feedback during real conditioning, like blocking the MB output during training.
- In the introduction and discussion sections, the authors conclude that their finding indicate a persistent recurrent activity (page 4: 107, page 16: 467) within the KC-to-pPAM network (page 16: 468) helping the stabilization of appetitive odor memory (page 4: 108, page 16:469), but I feel the evidence showed here could not sufficiently support their argument. I suggest the authors either provide more evidence or weaken/change their conclusions.
 - o None of their result could relate to a persistent activity in the pPAM or MB circuit. In order to support this idea, they would have to do functional imaging and show a persistently increased neural activity in this circuit.
 - o Direct KC-to-pPAM communication is also not fully approved in their results. They showed blocking pPAM function could impair the phenotype of MB activation, but this functional feedback could happen at MBON-to-pPAM or even downstream of MBON to pPAM. It is actually not a direct evidence for KC-to-pPAM feedback. This notion can also apply to the Imaging/optogenetics experiments, so the observation of elevated Calcium signal in pPAM after MB activation doesn't mean pPAM receives the direct feedback from MB. It could happen between the downstream of MB and pPAM.
 - o They showed the MB activation during normal odor-sugar training could induce a longer-lasting memory, but it doesn't mean the function of the feedback loop is to stabilize the memory. As I mentioned above, they would have to show the necessity of the feedback loop in memory consolidation, especially in a period between training and testing.

Minor concerns:

- Some important details are missing in the methods section:
 - o For optogenetics experiments, do they feed the flies with retinal or not? Although ChR2-XXI could work without retinal, it works much more efficiently with retinal. If not feeding retinal, do they use any

assay to test the ChR2-XXL is really functioning in their experiments?

o For olfactory preference tests, they didn't mention how long the flies were allowed to choose between odor and non-odor in each trail.

o For the functional imaging, they didn't describe the emission filter, and why do they need the image splitter and two sCMOS cameras in their experiment?

- The description of their statistic methods is also vague. I suggest them to also report the p-value, n number and statistic test for each experiment in their result section, and it will greatly help the audience to judge the results.

- For the comparison between different genotypes, why do they use a pairwise t-test instead of using ANOVA with proper post-hoc testing?

- Some labeling in the figure is confusing. I suggest the authors use ordinary labeling like figure 1A, 1B, 1C, instead of A, A', A''.

Reviewer #3:

Remarks to the Author:

This manuscript describes the functionality of a recurrent connection between Kenyon Cells (KCs) and PAM dopaminergic neurons (DANs) in *Drosophila*, which is important for appetitive learning and increases the persistence of appetitive memory. This type of recurrent connectivity was recently described for the aversive memory-encoding dopaminergic circuitry in adult flies. The manuscript represents an important contribution toward understanding how recurrent connectivity between KCs and dopaminergic neurons supports reward learning. Comments to improve the manuscript are below.

Major comments

- The technical approach is interesting, starting with broad activation of KCs. The manuscript would benefit if the authors laid out clearly, in the beginning of the results section, the rationale for stimulation KCs broadly. I was initially confused by this, and I suspect that nonspecialists will find it either completely impenetrable or draw an incorrect conclusion about what is being activated by the CS and what is being stimulated.

- The imaging experiments are important but not well flushed out. Time series traces should be shown before, during, and after stimulation with light. The reader should be able to see the traces return to baseline following stimulation, to rule out contribution of baseline shifts or other artifacts. Additionally, either all traces or mean +/- error must be shown (i.e., the authors should not select only the "exemplary responses"). Representative image(s) with ROIs should be included (main figure preferable, supplement if need be) to show the reader what was imaged.

- The observation that broad KC activation leads to appetitive memory is important and interesting. Two previous studies are directly relevant to the interpretation of these data and should be noted. Louis et al (2018 PNAS 115(3):E448-E457) reported that that appetitive conditioning (but not aversive), with naturalistic stimuli, increases relative KC responses to the CS+. This foreshadows the conclusion that activation of KCs induces appetitive memory and provides key support for the authors' model at the physiological level. In addition, Boto et al (Curr Biol 24(8):822-31) reported that reinforcement substitution pairing odor with thermogenetic PAM stimulation enhances activation of KCs across MB lobes.

- Overall, the present study represents a technically solid and important contribution to the field. There is conceptual overlap with Cervantes-Sandoval et al on the functionality of KC-DAN reciprocal

connectivity. This manuscript extends those findings into the appetitive realm and adds additional detail, though the novelty is borderline for this venue.

Minor comments

- Some of the detail in the introduction is superfluous and would likely be difficult to follow for nonspecialists. I would suggest streamlining it and removing any unnecessary connectome detail. OAN discussion should be removed, as is not particularly relevant and could be brought up in the discussion if the authors feel it is necessary. The bottom half of page 3 (lines ~86-96) comes off as rambling and awkward to read; most of it could be removed. The introduction would flow much better if the authors include only the connectomics data that is necessary to lay out the rationale for the present study (i.e., the reciprocal connections between KCs and DANs).

- I would suggest citing original literature rather than reviews, where possible. For instance, it would be possible, with a similar number of citations, to cite the early reinforcement substitution experiments at the top of page 3 (along with perhaps one major review, if necessary).

Reviewers' comments:

Reviewer #1 (Remarks to the Author):

The main claim of this work is that artificial and massive activation of mushroom body neurons induces appetitive memory through activating pPAM dopaminergic neurons. The finding is somewhat surprising, since small ensembles of these neurons have been well known to represent odors, and therefore the effect of massive activation could have disrupted odor specificity. The manuscript however suffers from the lack of evidence to prove that this recurrent circuit functions under physiological conditions.

1. All the data are based on massive activation of most if not all Kenyon cells. To the best of my knowledge, there is no evidence that this would happen under a natural condition. The authors must show supporting experimental evidence that the proposed circuit is functional during real learning. The authors could transiently block Kenyon cells during conditioning, downregulate acetylcholine receptors in the dopaminergic neurons, and/or monitoring sugar responses of Kenyon cells or the dopaminergic neurons upon Kenyon cell blockade.

→ *We thank the reviewer for his thoughts and agree. In previous work (Pauls et al. 2010) we provided evidence that a conditional block of KCs during training and test impairs odor-sugar memory formation. In the revised manuscript, we now provide more physiological data suggesting that a downregulation of ACh receptors in PAM neurons does not affect odor-sugar memories (which is different to published data on odor-shock memories in the adult fly; Cervantes-Sandoval et al. 2017). Interestingly, we found that downregulation of the sNPF receptor (the neuropeptide sNPF is known to be expressed in larval and adult KCs) impairs odor-sugar learning. This is further supported by data showing that (a) a knockdown of peptide processing specifically in KCs impairs the induction of "our" optogenetically-induced appetitive memory (H24>ChR2-XXL;amon-RNAi), (b) that pPAM neurons respond to bath applied sNPF during imaging using GCaMP6m in PAM cells, and (c) that pPAM neurons displayed a reduced Ca²⁺ response to optogenetically activated KCs when we in addition used amon-RNAi to knockdown peptide production. In the latter experiment, we used Chrimson instead of ChR2-XXL based on the presence of 5 different p-elements in these flies.*

Based on this novel mechanisms, we revised our main text and specified our conclusions.

2. The authors should tone down the interpretation of figure 5. Currently, there is no evidence showing the "sustained" activity of the recurrent mushroom body circuit.

→ *We completely agree with this concern and revised our manuscript with respect to data interpretation.*

3. Similarly, the necessity of the recurrent circuit in memory stabilization is unclear.

→ *We agree with this concern and revised our manuscript with respect to data interpretation.*

4. The authors should explain how odor identity is encoded under optogenetic activation of all Kenyon cells (Figure 2C). Is odor discrimination possible also with the other GAL4 driver (OK107)?

→ *Our data in figure 2 clearly shows that larvae are still able to discriminate odors. The focus of our paper is to dissect feedback signaling from KCs to dopaminergic pPAM neurons. Albeit it is certainly an interesting question "how" odor discrimination works though KCs are artificially activated, it is not our primary goal.*

5. How can figure 3C suggest that "optogenetic activation of KCs induces pPAM-dependent reward signaling (L270)"?

→ *We tried to clarify our thoughts about this experiment.*

Reviewer #2 (Remarks to the Author):

This manuscript by Lyutova et. al. tests an idea that a functional recurrent loop exists between Mushroom body Kenyon cells (KC) and a cluster of dopaminergic neurons (pPAM) in Drosophila larva brain. The authors first describe a novel phenomenon that artificial activation of KCs paired with odor cue is sufficient to trigger the appetitive memory, and this phenomenon is pPAM neurons dependent. Then, by using behavioral tests and functional imaging, they test if pPAM dopamine neurons mediate reward information to, and receive feedback signals from KCs. Last, the authors show optogenetic activation of KCs increases the persistence of odor-sugar memory. Based on those observations, they hypothesize that the function of the KC-to-pPAM recurrent loop and the sustained activity in this circuitry is to consolidate memories.

Overall, those observations are of interest and provide new insights for understanding the MB-DAN-MBON circuitry in Drosophila larva, but some conclusions still cannot be sufficiently supported by the current evidence in the manuscript.

Major concerns:

• In experiments for Fig. 1, two MB Gal4 lines are used, H24 and OK107, to confirm the involvement of Mushroom Body (MB). However, I would suggest them to further exclude the function of MB by combining H24 or OK107 with MB-Gal80.

→ *We thank the reviewer for this comment. We now provide data of two more Gal4 lines varying in the number of KCs. 201y-Gal4 (~315) and MB247-Gal4 (~341). With this, we have data using four different driver lines that only overlap in the mushroom bodies.*

- As they said, it is actually 'striking' to see the artificial activation of KCs could replace the sugar reward to induce an odor-specific appetitive memory, given that the odor identity of CS+ represented in the MB axons would be disrupted by the pan-MB activation. I suggest the authors to provide more results for backing up this key and unusual observation. For example, train the flies with one or two more odors to show the generality of this paradigm. They can also train the flies with one odor but test with a different odor to see if the training process cause a non-specific effect on odor-induced behavior. At very least, this odor identity issue should be well discussed in the discussion section.
- Although the authors claim the KC-substitution learning is sufficient to induce appetitive memory by using the two-odor reciprocal paradigm, but the memory score in fig. 2C is lower than one-odor paradigm and is not significantly different from one of its controls. I wonder their thoughts about the possible reasons?
 - *We can see the reviewer's confusion while comparing the two experiments and the concerns about odor specificity. However, the KC-substitution experiment using two odors was performed by one of the co-authors, and not as most other experiments by the first author. In the very beginning of the project we found that activation of sNPF-positive (using sNPF-Gal4) neurons elicits nociceptive responses in the larvae, and our initial approach was to use this optogenetically-induced response "as a US" during conditioning. However, although larvae showed nociceptive responses during illumination, sNPF>ChR2-XXL larvae showed an appetitive memory expression in the substitution experiment. Also here, we used the two-odor reciprocal design and can at least backup our data on H24>ChR2-XXL with an experiment using a different driver line. However, we decided not to include the pilot experiments using sNPF-Gal4 trying to keep a more streamlined manuscript.*
- For fig 4B', in the text, the authors claim experimental larvae showed no appetitive memory in contrast to the genetic control (page 11: 315), but the two groups seem no significant difference in the figure. Why is there both n.s. and ** signs for the two groups?
 - *In our learning experiments, it is most critical to compare learning performance between the experimental group and appropriate genetic controls. However, we also provide statistical analysis whether the performance is significantly different from chance level. In this experiment, optogenetic activation of KCs is sufficient to induce an appetitive memory in the control (**), but not in the experimental animals with knockdown of the DA receptor dDA1 (n.s.). However, performance was not different to each other based on the variation of data in the experimental group.*
- For the functional imaging with optogenetic activation experiments, since the baseline brightness of GCaMP is proportional to illumination power, I was confused why there is no change in the dF/F trace of the control group when the illumination power shift from low to high? Any normalization method used here? Also, for fig. 4C', the authors should compare the data within group but at different illumination level instead of comparing the two groups. The result for Chrimson activation is also confusing. The authors should use a low-level blue light for GCaMP imaging at the beginning of each trial and then add a red light to excite the chrimson leaving the GCaMP unaffected.
 - *We thank the reviewer to point on our insufficient description. Indeed, there is normalization to illumination power and therefore no obvious change in dF/F₀ after changing light intensity. We tried to clarify this in the Methods section. Further, we omitted Chrimson data from the Supplement and edited the imaging data presented in this figure.*
- For fig 5, the authors wanted to test the proposed KC-to-pPAM feedback loop affects normal appetitive olfactory learning, and propose its function in changing memory persistency. However, the evidence is not enough to support their assumptions, especially most evidence is based on artificial activation. It is possible that the memory created by artificial pan-MB activation is merely different in nature (stronger maybe) to the real sugar memory, so the observed memory curve can be explained by the co-existence of a strong and a weak memory traces created by artificial activation and real odor-sugar association. It will be more convincing to show some blocking experiments to unveil the function of MB feedback during real conditioning, like blocking the MB output during training.
 - *We thank the reviewer for this comment. Indeed, we missed to show that the artificial memory is potentially stronger than a sugar memory, and thus memory enhancement is not due to a synergistic effect. We now provide new data on "blue light-only" in addition to "sugar only" and "blue light+sugar". Still, we found that optogenetic activation of KCs + sugar exposure gives higher memory scores over time.*
- In the introduction and discussion sections, the authors conclude that their finding indicate a persistent recurrent activity (page 4: 107, page 16: 467) within the KC-to-pPAM network (page 16: 468) helping the stabilization of appetitive odor memory (page 4: 108, page 16:469), but I feel the evidence showed here could not sufficiently support their argument. I suggest the authors either provide more evidence or weaken/change their conclusions.
 - o None of their result could relate to a persistent activity in the pPAM or MB circuit. In order to support this idea, they would have to do functional imaging and show a persistently increased neural activity in this circuit.
 - o Direct KC-to-pPAM communication is also not fully approved in their results. They showed blocking pPAM function could impair the phenotype of MB activation, but this functional feedback could happen at MBON-to-pPAM or even downstream of MBON to pPAM. It is actually not a direct evidence for KC-to-pPAM feedback. This notion can also apply to the Imaging/optogenetics experiments, so the observation of elevated Calcium signal in pPAM after MB activation doesn't mean pPAM receives the direct feedback from MB. It could happen between the downstream of MB and pPAM.
 - o They showed the MB activation during normal odor-sugar training could induce a longer-lasting memory, but it doesn't mean the function of the feedback loop is to stabilize the memory. As I mentioned above, they would have to show the necessity of the feedback loop in memory consolidation, especially in a period between training and testing.
 - *We agree with the reviewer's concern. We now (a) tuned down our assumptions about persistent activity in the MB neuronal circuit. (b) In addition, we provide new data suggesting that a downregulation of ACh receptors in*

PAM neurons does not affect odor-sugar memories (which is different to published data on odor-shock memories in the adult fly; Cervantes-Sandoval et al. 2017). Interestingly, we found that downregulation of the sNPF receptor (the neuropeptide sNPF is known to be expressed in larval and adult KCs) impairs odor-sugar learning. This is further supported by data showing that (a) a knockdown of peptide processing specifically in KCs impairs the induction of "our" optogenetically-induced appetitive memory (H24>ChR2-XXL; amon-RNAi), (b) that pPAM neurons respond to bath applied sNPF during imaging using GCaMP6m in PAM cells, and (c) that pPAM neurons displayed a reduced Ca²⁺ response to optogenetically activated KCs when we in addition used amon-RNAi to knockdown peptide production.

Based on this novel mechanisms, we revised our main text and specified our conclusions.

Minor concerns:

• Some important details are missing in the methods section:

o For optogenetics experiments, do they feed the flies with retinal or not? Although ChR2-XXI could work without retinal, it works much more efficiently with retinal. If not feeding retinal, do they use any assay to test the ChR2-XXL is really functioning in their experiments?

→ In case for ChR2-XXL experiments we did not feed flies with additional retinal. As we induce a memory due to the function of ChR2 in KCs we assume there is no doubt that ChR2-XXL is functional. Further, we do see changes in intracellular Ca²⁺ levels in our imaging experiment which confirms ChR2-XXL to be functional. However, in case for Chrimson we added all-trans retinal to the food. Both is now mentioned in the Methods section.

o For olfactory preference tests, they didn't mention how long the flies were allowed to choose between odor and non-odor in each trail.

→ Similar to the test situation of learning experiments, larvae were tested after 3min. We apologize that this information was missing and added the information to the Methods section.

o For the functional imaging, they didn't describe the emission filter, and why do they need the image splitter and two sCMOS cameras in their experiment?

→ We apologize that this information was missing and added the information to the Methods section.

• The description of their statistic methods is also vague. I suggest them to also report the p-value, n number and statistic test for each experiment in their result section, and it will greatly help the audience to judge the results.

→ In each experiment the n-number is indicated below the boxplots (e.g. N=15). The statistical analysis is described in the Methods section and is constant throughout the experiments. To our point of view presenting a high number of p values within the main text is rather disturbing. However, we can provide a table with all p values in the supplementary – if requested.

• For the comparison between different genotypes, why do they use a pairwise t-test instead of using ANOVA with proper post-hoc testing?

→ As we by default have (mostly) three genotypes to compare, we usually start with testing for normal distribution with Shapiro Wilk test. Then, dependent on the outcome, we use a pairwise.t.test or pairwise.wilcox.test including a Bonferroni-Holm correction of test for significance. To our point of view this is a valid way for statistical analysis.

• Some labeling in the figure is confusing. I suggest the authors use ordinary labeling like figure 1A, 1B, 1C, instead of A, A', A''.

→ We agree and changed it accordingly.

Reviewer #3 (Remarks to the Author):

This manuscript describes the functionality of a recurrent connection between Kenyon Cells (KCs) and PAM dopaminergic neurons (DANs) in *Drosophila*, which is important for appetitive learning and increases the persistence of appetitive memory. This type of recurrent connectivity was recently described for the aversive memory-encoding dopaminergic circuitry in adult flies. The manuscript represents an important contribution toward understanding how recurrent connectivity between KCs and dopaminergic neurons supports reward learning. Comments to improve the manuscript are below.

Major comments

- The technical approach is interesting, starting with broad activation of KCs. The manuscript would benefit if the authors laid out clearly, in the beginning of the results section, the rationale for stimulation KCs broadly. I was initially confused by this, and I suspect that nonspecialists will find it either completely impenetrable or draw an incorrect conclusion about what is being activated by the CS and what is being stimulated.

→ We now provide data that an optogenetic activation of a smaller number of KCs (in 201y: 315 KCs and in MB247: 341 KCs) is also sufficient to induce the memory. As often the case, we accidentally found that activation of KCs is sufficient to induce a memory. We worked on the neuropeptide sNPF and its role in nociception. Activation of sNPF-positive neurons elicits nociceptive responses in the larvae, and our initial approach was to use this optogenetically-induced response "as a negative US" during conditioning. However, although larvae showed nociceptive responses during illumination, sNPF>ChR2-XXL larvae showed an appetitive memory expression. As the sNPF-Gal4 expresses in a high number of KCs, we used H24-Gal4 to check whether this effect is based on KC activation or sNPF-positive cells outside the MB present in sNPF-Gal4. In our opinion, it

would be rather confusing to tell this story in the paper to clarify why we started with H24-Gal4 which broadly expresses in KCs.

- The imaging experiments are important but not well flushed out. Time series traces should be shown before, during, and after stimulation with light. The reader should be able to see the traces return to baseline following stimulation, to rule out contribution of baseline shifts or other artifacts. Additionally, either all traces or mean +/- error must be shown (i.e., the authors should not select only the "exemplary responses"). Representative image(s) with ROIs should be included (main figure preferable, supplement if need be) to show the reader what was imaged.

→ We thank the reviewer for this comment. We used a normalization method to exclude pure baseline shifts and thus false positive data. However, we now present our imaging data in more detail, including mean traces + +/- errors.

- The observation that broad KC activation leads to appetitive memory is important and interesting. Two previous studies are directly relevant to the interpretation of these data and should be noted. Louis et al (2018 PNAS 115(3):E448-E457) reported that that appetitive conditioning (but not aversive), with naturalistic stimuli, increases relative KC responses to the CS+. This foreshadows the conclusion that activation of KCs induces appetitive memory and provides key support for the authors' model at the physiological level. In addition, Boto et al (Curr Biol 24(8):822-31) reported that reinforcement substitution pairing odor with thermogenetic PAM stimulation enhances activation of KCs across MB lobes.

- Overall, the present study represents a technically solid and important contribution to the field. There is conceptual overlap with Cervantes-Sandoval et al on the functionality of KC-DAN reciprocal connectivity. This manuscript extends those findings into the appetitive realm and adds additional detail, though the novelty is borderline for this venue.

→ Thanks a lot for the positive evaluation. We now provide more data and show that KC-to-pPAM feedback signaling is dependent on sNPF signaling rather than on acetylcholine (as shown for odor-shock learning in flies; Cervantes-Sandoval 2017). Thus, we provide a novel mechanism for feedback signaling crucial for reward memories in the Drosophila larva.

Minor comments

- Some of the detail in the introduction is superfluous and would likely be difficult to follow for nonspecialists. I would suggest streamlining it and removing any unnecessary connectome detail. OAN discussion should be removed, as is not particularly relevant and could be brought up in the discussion if the authors feel it is necessary. The bottom half of page 3 (lines ~86-96) comes off as rambling and awkward to read; most of it could be removed. The introduction would flow much better if the authors include only the connectomics data that is necessary to lay out the rationale for the present study (i.e., the reciprocal connections between KCs and DANs).

→ After careful read through, we agree with the reviewer's comment and revised our introduction.

- I would suggest citing original literature rather than reviews, where possible. For instance, it would be possible, with a similar number of citations, to cite the early reinforcement substitution experiments at the top of page 3 (along with perhaps one major review, if necessary).

→ We thank the reviewer for this comment. At top of page three we on purpose chose to cite reviews because our intention here was to summarize learning and memory in Drosophila over the years. In this part, the focus was not on substitution experiments, the term "activation of KCs" was probably confusing. Nevertheless, we tried to reduce the number of cited reviews throughout and revised the mentioned paragraph on page 3.

Reviewers' Comments:

Reviewer #1:

Remarks to the Author:

The authors provided additional evidence that validates their argument. They showed involvement of sNPF in the recurrent circuit between DAN and KC. These data together with the existing ones sufficiently make a case for their main claim. I list some minor comments to be improved before publication.

- About Fig. 1K', the result of their statistics seems strange. The double-asterisk between the right and left bars should be a mistake; looks as if it should be placed between the left and the center groups.

- Performance of H24>ChrR2-XXL flies is not significantly higher than one of their control flies' (Fig. 2I). Also, the authors did not provide evidence that differential conditioning is possible with other GAL4 drivers. Therefore, the claim that the KC-activation induced memory can support odor discrimination remains inconclusive. Considering that their main argument is not there, I suggest to drop this claim. Alternatively, they should perform new differential conditioning experiments with another driver.

Reviewer #2:

Remarks to the Author:

The authors have addressed all of my concerns.

Reviewer #3:

Remarks to the Author:

The present version of the manuscript is improved, though my major technical concern remains. Textual edits have improved the reading of the manuscript, but the GCaMP imaging experiments are unconvincing in the present version of the manuscript, and need major improvement if they are to be included in support of the authors' model (and they are important).

Optogenetically stimulating KCs should produce a rapid rise in Ca²⁺ in postsynaptically-connected neurons that falls back to baseline when the stimulation is ended (notwithstanding the nonconventional stimulation/imaging protocol). This is not shown in figure 4G (and others). The data shown – long periods of ChrR2 stimulation with traces changing slowing over 5-min and no return to baseline tested/shown – are not very convincing. As noted in the previous round of review, the authors need to provide: A) representative images showing the nervous system with the ROIs they imaged, as well as B) traces before, during, and after stimulation (not just during), which should include C) either all individual traces or the mean +/- S.E.M. An unspecified normalization procedure is not sufficient to alleviate this concern. It seems that the authors may be showing only the stimulation period due to their protocol of changing blue light levels to stimulate Chr2 while simultaneously imaging with [low-intensity] blue light. Regardless of the rationale, though, this issue needs to be addressed, the data shown, and the methodology clearly described. How the authors do so is up to them (one can think of multiple possibilities); however, the best option would be to spectrally separate their stimulation and imaging, running an experiment with ChrR2/RCaMP1a or GCaMP/CsChrimson combinations. This would allow them to show complete time series traces that would, assuming their hypothesis is correct, provide more convincing support.

Reviewers' comments:

Reviewer #1 (Remarks to the Author):

The authors provided additional evidence that validates their argument. They showed involvement of sNPF in the recurrent circuit between DAN and KC. These data together with the existing ones sufficiently make a case for their main claim. I list some minor comments to be improved before publication.

- About Fig. 1K', the result of their statistics seems strange. The double-asterisk between the right and left bars should be a mistake; looks as if it should be placed between the left and the center groups.

→ *We thank the reviewer for drawing attention to this mistake. We indeed switched the position of double-asterisk and the "n.s.". We corrected this in the current version of the manuscript.*

- Performance of H24>Chr2-XXL flies is not significantly higher than one of their control flies' (Fig. 2I). Also, the authors did not provide evidence that differential conditioning is possible with other GAL4 drivers. Therefore, the claim that the KC-activation induced memory can support odor discrimination remains inconclusive. Considering that their main argument is not there, I suggest to drop this claim. Alternatively, they should perform new differential conditioning experiments with another driver.

→ *We understand the reviewer's concern. We now added a second experiment and this time used another driver line (Ok107-Gal4) in the two-odor reciprocal experiment (Fig. S6, S6'). Again, experimental larvae showed appetitive memory expression, this time significantly different to both genetic controls. Consequently, larvae must be able to distinguish between the two odors used (AM and OCT) despite the activation of KCs by Chr2-XXL. We changed the manuscript accordingly in line 185-191.*

Reviewer #2 (Remarks to the Author):

The authors have addressed all of my concerns.

→ *Thank you for your efforts and time.*

Reviewer #3 (Remarks to the Author):

The present version of the manuscript is improved, though my major technical concern remains. Textual edits have improved the reading of the manuscript, but the GCaMP imaging experiments are unconvincing in the present version of the manuscript, and need major improvement if they are to be included in support of the authors' model (and they are important).

Optogenetically stimulating KCs should produce a rapid rise in Ca²⁺ in postsynaptically-connected neurons that falls back to baseline when the stimulation is ended (notwithstanding the nonconventional stimulation/imaging protocol). This is not shown in figure 4G (and others). The data shown – long periods of Chr2 stimulation with traces changing slowing over 5-min and no return to baseline tested/shown – are not very convincing. As noted in the previous round of review, the authors need to provide: A) representative images showing the nervous system with the ROIs they imaged, as well as B) traces before, during, and after stimulation (not just during), which should include C) either all individual traces or the mean +/- S.E.M. An unspecified normalization procedure is not sufficient to alleviate this concern. It seems that the authors may be showing only the stimulation period due to their protocol of changing blue light levels to stimulate Chr2 while simultaneously imaging with [low-intensity] blue light. Regardless of the rationale, though, this issue needs to be addressed, the data shown, and the methodology clearly described. How the authors do so is up to them (one can think of multiple possibilities); however, the best option would be to spectrally separate their stimulation and imaging, running an experiment with Chr2/RCaMP1a or GCaMP/CsChrimson combinations. This would allow them to show complete time series traces that would, assuming their hypothesis is correct, provide more convincing support.

→ *We do see the concerns of the reviewer here and therefore changed our imaging protocol. We now added a filter to significantly decrease our light intensity used for GCaMP imaging (in the previous version we used light intensity of about 700 μW/cm², now we used 500 μW/cm²). We still see an increase in calcium concentration/fluorescence intensity directly in response to light, but we now are able to come back to a stable baseline after the stimulation of Chr2-XXL (see Fig. 4G, S8-S10). Still, we have to state that we use the same wavelength for imaging GCaMP and activating Chr2-XXL (the same for Chrimson in Fig. S11-S13) and therefore we do see an increase in calcium concentration/fluorescence intensity also in the first part of the experiment (0-300s; Fig. S8; as shown in the previous version of the experiment). But now, after the first light pulse - while going back to the lower light intensity - we see a falls back to baseline and a repeated increase in calcium concentration/fluorescence intensity due to a second increase in light intensity.*

In detail, we now used the following protocol (we introduced the changed protocol in the material and methods section in the current version of the manuscript, line 948-966):

We dissected the brain under red light and let it rest in HL3.1 in complete darkness for at least 30 minutes. Then we started imaging (80ms exposure time; 500 $\mu\text{W}/\text{cm}^2$ light intensity; 475nm; 2s time interval) and after 300s (150 time points) we paused imaging to give a 1 minute continuous light pulse (1000 $\mu\text{W}/\text{cm}^2$ light intensity; 475nm). We resumed imaging and after 960s (480 time points) we again gave a continuous 1minute light pulse of higher light intensity (1000 $\mu\text{W}/\text{cm}^2$ light intensity; 475nm). Afterwards we continued to image for at least 240s (120 time points) with 500 $\mu\text{W}/\text{cm}^2$ low light intensity.

With this protocol we believe to reduce the activation of ChR2-XXL already with the light necessary to image GCaMP. After the first light pulse (after 300s) we do see a rapid increase of calcium concentration in the pPAM neurons, but we also see a decrease in calcium concentration back to baseline afterwards, while the second light pulse increases the calcium concentration again. Also here we do see the falls back to baseline. We show the mean \pm SEM traces including 120s before and after the second light pulse for the activation experiment using ChR2-XXL in KCs and GCaMP6m in DANs (Fig. 4G-I) and the individual traces and the complete imaging traces (601s) in the supplement (S8-S10). Additionally, we now added a representative picture of the ROIs we chose (S10).

Beside changes in the material and methods section (line 948-966), the manuscript is changed accordingly to the new imaging data in line 309-317. New imaging data is presented in Figure 4G-I and Supplement Figure S8-S10.

Accordingly, we changed our imaging protocol for challenging calcium concentration/fluorescence intensity in in pPAM neurons in response to activation of KCs while using amon-RNAi to knock down peptide processing in KCs (Figure S11-13).

Reviewers' Comments:

Reviewer #1:

Remarks to the Author:

All my points have been appropriately addressed.

Reviewer #3:

Remarks to the Author:

The new data/experiments have sufficiently addressed my concerns and significantly improved the manuscript. I have no further concerns.

REVIEWERS' COMMENTS:

Reviewer #1 (Remarks to the Author):

All my points have been appropriately addressed.

→ *Thanks a lot for your time and efforts.*

Reviewer #3 (Remarks to the Author):

The new data/experiments have sufficiently addressed my concerns and significantly improved the manuscript. I have no further concerns.

→ *Thanks a lot for your time and efforts.*